# Measurement Report: A survey of meteorological and cloud properties during ACTIVATE's postfrontal flights and their suitability for Lagrangian case studies

Florian Tornow[1,2], Ann Fridlind[2], George Tselioudis[2], Brian Cairns[2], Andrew Ackerman[2], Seethala Chellappan[3,4], David Painemal[3,4], Paquita Zuidema[5], Christiane Voigt[6,7], Simon Kirschler[6], and Armin Sorooshian[8]

[1]Center for Climate Systems Research, Columbia University
[2]NASA GISS
[3]NASA LaRC
[4]Analytical Mechanics Associates, Hampton, VA
[5]Department of Atmospheric Sciences, Rosenstiel School, University of Miami
[6]Institute of Atmospheric Physics, Deutsches Zentrum für Luft und Raumfahrt (DLR), Oberpfaffenhofen, Germany
[7]Institute of Atmospheric Physics, Johannes Gutenberg-University, Mainz, Germany
[8]Department of Chemical and Environmental Engineering, University of Arizona

**Correspondence:** Florian Tornow (ft2544@columbia.edu)

**Abstract.** Postfrontal clouds, often appearing as marine cold-air outbreaks (MCAOs) along Eastern seaboards, undergo overcast-to-broken cloud regime transitions. Earth system models exhibit diverse radiative biases connected to postfrontal clouds, rendering these marine boundary layer (MBL) clouds a major source of uncertainty in projected global-mean temperature. The recent NASA multi-year campaign Aerosol Cloud meTeorology Interactions oVer the western ATlantic Experiment (ACTIVATE) therefore dedicated most of its resources to sampling postfrontal MCAOs, deploying 71 flights from 2020 through 2022. We provide an overview of (1) the synoptic context within the parent extratropical cyclone, (2) the meteorological conditions with respect to the season, (3) the suitability of case data and measurements for Lagrangian analysis and modeling studies, and (4) the encountered cloud properties. A proposed subset of flights deemed most suitable for Lagrangian modeling case studies is highlighted throughout. Such flights typically cover a greater fetch range, were better aligned with the MBL wind direction, and revisited sampled air masses, when key instruments were operational. Like many other flights, these flights often probed cloud formation and some cloud regime transitions. Surveying cloud properties from remote sensing and in-situ probes, we find a great range in cloud-top heights and a relatively large concentration of frozen hydrometeors, which suggest strong free tropospheric entrainment and secondary ice formation, respectively. Both processes are expected to leave marked signatures in cloud evolution, such as strongly ranging cloud droplet number concentrations. ACTIVATE data combined with satellite retrievals can establish observational constraints for future model improvement work.

# 1 Introduction

Postfrontal low-level clouds occur frequently over the extratropical oceans globally. Their relatively poor representation in earth system models (ESMs, e.g., Bodas-Salcedo et al., 2014) and undetermined cloud-climate feedback (e.g., Frey and Kay, 2018; McCoy et al., 2019; Zelinka et al., 2022; McCoy et al., 2023) substantially contribute to the uncertainty in global mean temperature projections (Bock et al., 2020; Zelinka et al., 2020). Thus, postfrontal clouds emerge as an important target for field campaigns and subsequent model-observation intercomparison to test and improve ESMs. Efforts to improve ESM cloud physics have often benefited from side-by-side comparison of large-eddy simulations and ESM simulations in single column model (SCM) mode (e.g., Neggers, 2015). The SCM simulations efficiently express parent ESM column physics skill and bias while being forced by well-defined boundary conditions, which typically reflect synoptic conditions leading to problematic cloud representation, hereafter referred to as modeling case studies. Where horizontal advection is large (e.g., in postfrontal situations), these boundary conditions can be extracted along Lagrangian trajectories that follow the cloudy air mass, thereby enabling simulations of a horizontally translating domain that are locally free of leading horizontal advective tendencies (e.g., Neggers, 2015; Pithan et al., 2019).

Postfrontal clouds, often appearing as marine cold-air outbreaks (MCAOs), are challenging to represent in ESMs (Pithan et al., 2019). After the cold front of an extratropical cyclone passes continental eastern coastlines, including the US eastern seaboard, a northwesterly flow of stronger wind speed typically sets in, transporting relatively cold air over a relatively warm ocean surface and spurring intense turbulent surface fluxes (e.g., Painemal et al., 2023). The subsiding motion in the free troposphere (FT) aloft, often associated with dry intrusions (Browning, 1997; Raveh-Rubin, 2017), creates a capping inversion under which marine boundary layer (MBL) clouds begin forming at some distance downwind from the coastline. In MCAOs, these initial MBL clouds often appear as cloud streets (Brümmer, 1999). After filling in towards a nearly or fully overcast cloud deck within the MBL farther downwind, clouds then transition towards a broken, sometimes open-cellular deck at greater fetch. There is evidence that the different morphologies (i.e., overcast versus broken clouds) coincide with distinct meteorological boundary conditions (McCoy et al., 2017; Chen et al., 2022). A key driver of the cloud regime transitions is often the formation of substantial precipitation (e.g., Abel et al., 2017; Tornow et al., 2023; Seethala et al., 2024; Kirschler et al., 2023). Hydrometeor collisions, leading to coalescence of droplets, droplet collection by rain drops, and droplet collection by frozen hydrometeors (riming), efficiently reduce the number concentration of cloud droplets and thereby aerosol available as cloud condensation nuclei (CCN) and thereby the number concentration of cloud droplets. Below-cloud scavenging of aerosol can further remove CCN. Similar to drizzle-driven transitions in the subtropics (Yamaguchi et al., 2017), the loss of CCN amplifies subsequent precipitation formation, creating a positive feedback loop that is especially efficient where cloud condensate reaches high mixing ratios (Wood et al., 2017). With intensifying precipitation, sub-cloud evaporation of precipitation progressively stratifies the MBL, inhibiting vertical transport of heat, moisture, and CCN (often referred to as 'decoupling', e.g., Abel et al., 2017; Yamaguchi et al., 2017), thereby transforming a stratiform-natured cloud deck into a convective-natured one (Field et al., 2014; Tomassini et al., 2017) and presenting challenges for ESM physics in representing the coupled microphysical and dynamical processes (Pithan et al., 2019).

Aircraft and surface-based campaigns provide crucial measurements to improve our understanding of regional MCAO cloud regimes. For example, GALE (Genesis of Atlantic Lows Experiment, Dirks et al., 1988) probed extratropical cyclones and MCAO clouds over the NW Atlantic. NAAMES (North Atlantic Aerosols and Marine Ecosystems Study, Behrenfeld et al., 2019) probed MCAOs farther north, illuminating ocean and meteorological processes modifying size and mass composition with fetch (e.g., Sanchez et al., 2018). A more comprehensive overview for the NW Atlantic is provided by Sorooshian et al. (2020). Much farther downwind, the ACE-ENA campaign (Aerosol and Cloud Experiments in the Eastern North Atlantic, Wang et al., 2019) sampled sporadic postfrontal passages, often after cloud regime transitions occurred. Over the Norwegian Sea, COMBLE (Cold-Air Outbreaks in the Marine Boundary Layer Experiment, Geerts et al., 2022) similarly sampled MCAOs from a surface site ∼1000 km downwind of MCAO inception and connected to an upwind site near inception (e.g., Williams et al., 2024), enabling a model-observation intercomparison focused on aerosol-cloud interactions in sub-Arctic MCAOs (Juliano et al., 2024). Additional MCAO data over the Norwegian Sea have now been gathered during the recent CAESAR (Cold Air Outbreak Experiment in the Sub-Arctic Region) campaign. A comprehensive set of field campaigns focused on MCAO in northern latitudes, including flight campaigns with bases in Spitsbergen, Norway and Kiruna, Sweden, for example (AC)[3] (Wendisch et al., 2023) that included AFLUX (Aircraft campaign observing FLUXes of energy and momentum in the cloudy boundary layer over polar sea ice and ocean), MOSAiC-ACA (Multidisciplinary Drifting Observatory for the Study of Arctic Climate – Airborne observations in the Central Arctic, Mech et al., 2022; Moser et al., 2023), and HALO-(AC)[3] (High Altitude and Long Range Research Aircraft – AC3 project, Wendisch et al., 2024).

This study focuses on postfrontal MCAO flights during the recently concluded multi-year NASA Earth Venture Suborbital (EVS) campaign ACTIVATE (Aerosol Cloud meTeorology Interactions oVer the western ATlantic Experiment, Sorooshian et al., 2019). ACTIVATE deployed two aircraft that flew in tandem most of the time (Schlosser et al., 2024) and carried advanced instrumentation (Sorooshian et al., 2023): (1) a high-flying King Air equipped with remote sensing instruments and dropsondes, and (2) a low-flying Falcon that porpoised through the MBL and FT, comprehensively measuring aerosol and cloud properties using in-situ probes. The base of operations for almost all ACTIVATE flights was NASA Langley Research Center (LaRC) in Hampton, Virginia, with flights typically being 3-4 hours with a subset of days with more desirable conditions having two tandem flights on the same day. Between the years 2020 and 2022 a total of 162 joint flights took place (Sorooshian et al., 2023), of which 71 flights show maximum marine cold air outbreak indices (here defined as $M = \theta_{\mathrm{srf}} - \theta_{850\,\mathrm{hPa}}$) greater than zero and are further examined here. An advantage of the multi-year deployment is the ability to perform repeated measurements during a specific season over multiple years, thereby building substantial statistics of key properties. To date, a few selected MCAOs have been targeted, but a comprehensive overview has not been done. Li et al. (2021) examined 28 February and 1 March 2020 using Eulerian LES and explored the dependence on meteorological forcing. With a focus on 1 March 2020, Chen et al. (2022) investigated the mesoscale cloud morphology in mesoscale simulations, while Tornow et al. (2022) studied aerosol dilution from FT entrainment. Seethala et al. (2024) surveyed numerous flights (i.e., 1 March 2020, 29 January 2021, 3 February 2021, 5 March 2021, and 8 March 2021) to explore mixed-phase cloud microphysical properties with distance from the coast.

This paper aims for (1) an overview of the synoptic conditions and the associated meteorological properties during ACTIVATE's postfrontal flights, (2) an assessment of the eligibility of each flight to align well enough with the prevalent wind direction and cover a wide enough fetch range, among others, to support a Lagrangian analysis or modeling case study, and (3) a survey of encountered cloud properties. We propose a subset of flights as most suitable for Lagrangian case studies, and survey that subset compared with all flights and the postfrontal class as a whole. The manuscript is organized as follows: Section 2 describes the data and methodology, Section 3 contains the overview analysis, Section 4 discusses the results, and Section 5 provides conclusions.

## 2 Data and Methods

### 2.1 Identification of cold front and low-pressure center locations

To identify cold front locations, we apply two techniques and thereby broadly follow earlier studies (e.g., Naud et al., 2016) in (1) searching for strong spatial features in potential temperature fields (Hewson, 1998), and (2) searching for strong temporal changes in wind speed and direction (Simmonds et al., 2012). We explain both methods in more detail below. For meteorological fields, we rely on the MERRA-2 reanalysis (Gelaro et al., 2017), which has been extensively compared to ACTIVATE data (Seethala et al., 2021). A resulting frontal location can be seen in Fig. 1. We note that MERRA-2 and other reanalyses are expected to locate widespread and long-lived mid-latitude post-frontal sectors quite accurately spatiotemporally. However, we consider MERRA-2 to less reliably predict quantities such as liquid and ice water path and associated cloud cover and albedo (Pithan et al., 2019). This motivates use of reanalysis fields to contextualize ACTIVATE aircraft data for improving understanding of MCAO microphysics and radiative impacts.

First, we use MERRA-2 potential temperature at 850 hPa, $\theta$, spatially smooth fields by computing a moving-window average using a 3×3 grid window (translating into a $1.5° \times 1.875°$ Latitude-Longitude box), and then calculate spatial derivatives, following Hewson (1998) and updates provided in Berry et al. (2011): a first-order derivative ($\frac{d\theta}{d[x/y]}$) signifying temperature change in longitudinal as well as latitudinal direction and second- and third-order derivatives of absolute temperature changes ($\frac{d}{dx}|\frac{d\theta}{dx}|$ and $\frac{d^2}{dx^2}|\frac{d\theta}{dx}|$). In addition, we compute the along-wind divergence using MERRA-2 850 hPa horizontal winds, $\boldsymbol{v_h}$, and above second-order derivatives ($\boldsymbol{v_h}\frac{d}{dx}|\frac{d\theta}{dx}|$). After excluding points within temperature dips (i.e., in areas where first-order and second-order derivatives align in sign) and where the product of first- and normalized second-order derivatives are small (excluding values above a threshold of $0.5e^{-10}$ K m$^{-1}$), we select points near third-order derivatives that are zero (i.e., extreme thermal gradients) and apply a threshold to filter for small along-wind divergence (values below 0.75 m s$^{-1}$). Lastly, we connect filtered points towards lines by looping over all points and searching within 250 km for neighboring members. Lines that exceed a maximum member-to-member distance of 250 km are considered as fronts. Where available we filter for line length greater than 500 and 1000 km.

Second, following Simmonds et al. (2012), we extract MERRA-2 fields at time steps three hours before and after the time of interest. We filter for grid points of strong 850 hPa meridional wind speed changes ($\frac{d|\boldsymbol{v_h}|}{dt} > 1/3$ m s$^{-1}$ h$^{-1}$), and change in

meridional wind direction (a switch in sign of $v_h$ component). Again, we connect filtered points as lines by looping over all points and searching within 250 km for neighboring members.

Lastly, we locate the low-pressure center as a simple minimum in surface pressure, limiting ourselves to the NW Atlantic domain (Lat. < 55 °N and Lon. < -40 °E).

## 2.2 Lagrangian trajectories

We construct underlying MBL Lagrangian trajectories at every 10 minutes along each track flown by the Falcon aircraft. Using MERRA-2 three-dimensional wind fields ("inst3_3d_asm_Nv") at the timestep closest to the time of interest, we interpolate horizontal wind components at a given start location and an altitude of 250 m, which we assume is representative of the MBL (e.g., Seethala et al., 2021). Using these winds, we then compute the expected position at the next time step (i.e., after three hours) and iterate the above procedure for 30 hours into the future and also 15 hours into the past, totaling a 48 hour trajectory. Along each trajectory we collocate MERRA-2 surface pressure, sea-surface temperature (SST), sensible and latent heat fluxes, and profiles of horizontal and large-scale vertical wind, relative humidity, and water vapor and cloud condensate mixing ratios. We use SST to limit trajectories to portions over the ocean.

## 2.3 GOES-16 SatCorps retrieval

To each of the MBL Lagrangian trajectories we collocate GOES-16 cloud retrievals that are typically available every 20 minutes. We approximate the location along the trajectory by interpolating latitude and longitude at acquisition time. We collect cloud retrievals within a window of $\pm 50$ km cross-track and $\pm 25$ km along-track direction, effectively forming a wind-oriented box of $100 \times 50$ km$^2$.

Cloud optical depth (COD) is retrieved during day- and nighttime. Daytime COD is primarily derived from the 0.64-$\mu$m channel, whereas nighttime COD (solar zenith angle > 82.5°) is estimated from 3 infrared channels (Minnis and Heck, 2012). The nighttime physical algorithm is only sensitive to clouds with COD < 6.0, and for those clouds, the retrievals compare well against independent observations (Minnis and Heck, 2012). For COD > 6.0, the algorithm is unable to discern the exact COD magnitude, and thus, these values should only be used for qualitative purposes (e.g. identification of optically thick clouds). Lastly, we determine cloud cover within the box by computing the portion of pixels with COD > 2.5 (Wyant et al., 1997). Retrievals are provided at a 4 km by 4 km resolution at nadir. While including clouds of all heights, we verify that clouds are mostly of low-level character (see Section 2.5). Fig. A1 shows $\sim$75 % of cloud-top heights within 3.5 km of the surface and $\sim$15 % above 5.0 km.

## 2.4 Selected ACTIVATE remote sensing and in-situ measurements

In addition to the flight track (i.e., using the Falcon aircraft location and timestamp unless stated otherwise), we collect measurements from selected remote sensing instruments and in-situ probes aboard the King Air and the Falcon, respectively:

- **SPEC Fast Cloud Droplet Probe (FCDP, Knop et al., 2021; Kirschler et al., 2022)** measures aerosol and cloud particles in diameter size range of 3-50 $\mu$m, sorted into 18 size bins, and reports $N_d$ as sum of all bins, as well as the particle number concentration in each bin,

- **SPEC 2D Stereo Probe (2DS, Lawson et al., 2006; Kirschler et al., 2023)** covers hydrometeors in the diameter size range of 11.4-1465 $\mu$m, sorted into 128 size bins. Particle size distributions are derived from FCDP for particles $< 30$ $\mu$m and from 2D-S for particles $> 30$ $\mu$m, and particles $< 100$ $\mu$m are assumed to be liquid droplets, as no other information was available (Kirschler et al., 2023). Hydrometeors greater than 100 $\mu$m are classified into liquid or frozen ice phase by the shape of the 2DS image and reported as the sum of all frozen hydrometeors $N_i$. The classification algorithm conservatively labeled hydrometeors as liquid when imagery was ambiguous, leading to a small number ($< 1\%$) of false positives for frozen hydrometeors but sporadically elevated ($< 40$ %) false positives for liquid hydrometeors,

- **Condensation Particle Counters TSI CPC-3776 and CPC-3772** measured condensation nuclei (CN) concentrations greater than 3 and 10 nm, respectively,

- **Research Scanning Polarimeter (RSP, Cairns et al., 1999)** performed passive polarimetric cloud remote sensing when sun-object-observer geometries were such that scattering angles from 135-155°, where the cloud bow is located, were observable. The location and structure of the cloud bow provides detailed information about the cloud top droplet size distribution. During winter and spring deployments this meant that the angle between the aircraft heading and the bearing of the sun had to be between 10 and 20°.

- **High Spectral Resolution Lidar 2 (HSRL-2, Burton et al., 2018)** performed active temperature, aerosol, and cloud remote sensing whenever located over ocean, measuring backscatter signals at 355 and 1064 nm, allowing, among others, to retrieve cloud-top height and cloud-top temperature; the latter obtained from vertical temperature profiles at the altitude of the former.

King Air measurements were collocated to Falcon ones via their nearest timestamp. In most cases the aircraft were distanced less than 6 km and within 5 min (Schlosser et al., 2024). The reader is referred to Sorooshian et al. (2023) for more details about the instrumentation and flight strategy details.

## 2.5 Evaluation criteria to assess combined case and flight qualities

Per ACTIVATE flight we verify a range of criteria (listed in Table 1 and also shown in Fig. 4, left). Criteria are derived from ACTIVATE in-situ and remote sensing data, Lagrangian trajectories, and collocated GOES-16 cloud cover. These criteria are designed to indicate several qualities:

(1) **Stereotypical postfrontal conditions** that often emerge as MCAOs. We filter for an Eastward boundary layer wind direction ('Primarily Eastward flow') that is expected from extratropical cyclone dynamics in the postfrontal sector (e.g. Tselioudis and Grise, 2020). To obtain strong MCAOs that are expected to undergo faster cloud regime transitions in

better reach of the aircraft during ACTIVATE., we also impose a MCAO index threshold ('Maximum MCAO index > 10 K'), which we also consider a proxy for elevated surface fluxes (and were therefore left out as criterion). Typically prevalent large-scale subsidence should disallow high-level clouds that may hinder satellite retrievals; we examine GOES-16 retrievals to verify the absence of high-level clouds ('High cloud fraction < 10 %').

(2) **Flights that followed the MBL air mass in a quasi-Lagrangian and also Lagrangian manner with key instruments being operational.** We use trajectory and flight path locations to measure spatial alignment, rewarding alignment during a sizeable fraction of the flight ('Downwind portion with respect to flight > 20 %') and additionally over a certain distance ('Downwind distance > 200 km') that we consider large enough to detect cloud property changes. We further examine if RSP, a key instrument to measure cloud micro- and macrophyiscal properties, was able to probe a sizeable portion ('Downwind RSP availability > 30 %') across this distance. Lastly, we explore the availability of Lagrangian airmass revisits, ensuring that horizontal legs are long enough to have aircraft probes collect data ('Minimum duration of Lagrangian legs > 1 minute'). We apply a similar metric to RSP as well ('Lagrangian RSP availability') to ascertain favorable potential sun-object-observer geometries.

(3) Where possible **liquid and frozen precipitation-sized hydrometeors that may drive the larger cloud regime transitions**, as well as **elevated aerosol concentrations from new particle formation that may delay transitions**. We use data from in-situ cloud probes to indicate the presence of the various hydrometeor types ('Frozen hydrometeors measured', 'Drizzle-sized particles measured' and 'Rain-size particles measured'). We additionally survey in-situ aerosol probes for traces of new particle formation; we use condensation nucleus ratios to indicate the presence of freshly nucleated particles 3 nm < $D_p$ < 10 nm ('Recent particle formation frequency > 5 %', e.g., Corral et al., 2022) that are expected to grow into larger sizes and drastically increased concentrations ('High aerosol concentration frequency > 5 %').

The criteria can be readily modified in the archived code base but are currently set to verify at least a few times across all flights. For example, selecting a radius exceeding 20 km for Downwind portion with respect to flight > 20% would increase the chance for this criterion to verify, while increasing the fetch range in Downwind distance > 200 km would diminish chances.

## 3 Results

### 3.1 Large-scale Cyclonic Context

From all ACTIVATE tandem flights coinciding with positive MCAO indices, we first locate features of the parent extratropical cyclone that allow us to put flight data into a larger meteorological context. From coinciding MERRA-2 meteorological fields we identify the low-pressure center and the cold front location. Fig. 1 shows an example from 29 March 2022 via satellite imagery and meteorological fields closest to the morning flight (thick yellow line). The two frontal identification methods (Sec. 2.1) are somewhat complementary and line up with frontal clouds in geostationary imagery:

1. searching for spatial gradients in temperature and wind fields and connecting clusters of strong gradients (Hewson, 1998) typically finds areas of greater baroclinicity and results here in elongated structures farther away from the low, and

2. searching for grid boxes that experienced above-threshold temporal changes in wind speed and direction (Simmonds et al., 2012) and connecting greater clusters typically locates areas of smaller baroclinicity, resulting in regions near the low-pressure center.

Uncertainties are expected where MERRA-2's three-hourly resolution fails to resolve fast-translating portions of the front, for example between -72° and -60° E (i.e., at the bottom of the Fig. 1) that is identified as farther west and disconnected from its north-eastern extension, contrary to satellite imagery.

Fig. 2 summarizes all flights by their surface pressure differences to the nearest identified cold front (x-axis) and the low-pressure center (y-axis), with positive differences typically indicating a position to the west of the cold front and to the south of the low, respectively. ACTIVATE flights cover a broad range within postfrontal areas, in particular where pressure differences between flight portion and cold front as well as the low fall between 0-20 hPa as well as 10-60 hPa, respectively. In a few instances, the ACTIVATE flights reached the cold front (i.e., a pressure difference to cold front smaller 0 hPa). Color shading in Fig. 2 panels shows reanalysis-based MBL horizontal and FT large-scale vertical wind speed of all flight points. Both panels reveal a general tendency for increased MBL wind speed and more positive vertical motion (with few incidents of lofting motion) closer to the low and the front, broadly matching earlier extratropical cyclone composites from satellite (Field and Wood, 2007; Naud et al., 2016) and in line with the general schematic of a heterogeneously subsiding dry intrusion (Browning, 1997).

## 3.2 Comparison to Seasonal Meteorology

ACTIVATE targeted postfrontal conditions, thereby avoiding other meteorological regimes (e.g., frontal passage with vertically extended clouds or regimes dominated by high surface pressure). To put the meteorological properties during flights into a greater seasonal context, we extracted MERRA-2 fields averaged over the campaign domain (i.e. a triangle spanning 32.50° N 80.00° W, 32.50° N 65.00° W, and 40.50° N 72.25° W). For the season November 2021 through March 2022, Fig. 3 shows timelines for selected meteorological parameters as well as their histogram (shown as box-whisker plots on the far right). In addition to the overall season and dates during postfrontal ACTIVATE flights, we also show a proposed subset of flights that may provide valuable Lagrangian case studies for analyses and modeling work, as analyzed in greater detail in Sec. 3.3. Fig. 1 depicts one of the flights within this subset.

As best seen in the timeline of surface fluxes (Fig. 3 bottom), postfrontal conditions occurred regularly, as often as every 3-4 days (e.g., in January 2022), and ACTIVATE probed about a third of all events. Compared to the greater season over the campaign domain (blue), ACTIVATE flights (grey) generally coincided with greater subsiding motion, roughly similar wind speeds, and considerably greater MCAO indices that also resulted in considerably larger surface fluxes. The subset of flights with most Lagrangian sampling during ideal postfrontal conditions (red) tends to represent relatively strong MCAOs, as evidenced by interquartile range (IQR) values above the IQR of all postfrontal flights as well as the overall season (shown

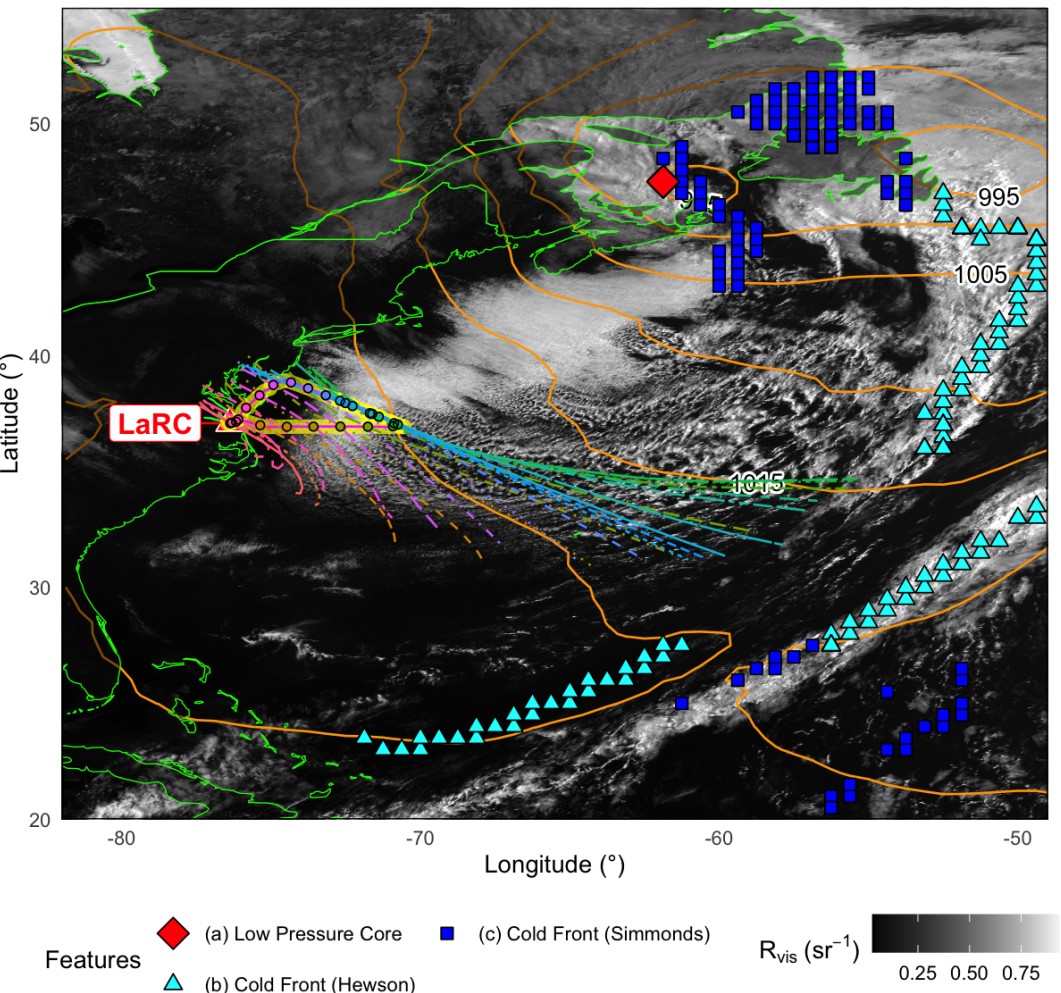

**Figure 1.** For a marine cold-air outbreak during the second flight on 29 March 2022, we demonstrate the identification of the low-pressure center (red diamond) as well as cold front locations using two approaches (cyan and blue symbols, Section 2.1). Along the flight track (yellow line), we launch Lagrangian back- and forward trajectories every 10 minutes (dots along track and lines of same color), clipped to only cover ocean surfaces (country borders shown in green lines). The background shows coinciding GOES-16 visible imagery (black-white scale) and MERRA-2 surface pressure (orange lines) corresponding to the midpoint of the flight time. The red-white triangle labeled with "LaRC" marks the airbase in Hampton, Virginia.

from top to bottom in Fig. 3): vertical motion is more negative (median of -25.0 mm s$^{-1}$ compared to -8.4 and -1.0 mm s$^{-1}$ for all flights and the overall season, respectively), wind speed is greater (median of 10.6 m s$^{-1}$ compared to 6.6 and 8.2 m s$^{-1}$, respectively), and MCAO indices (median of 7.4 K compared to 3.0 and -1.8 K, respectively) and resulting turbulent

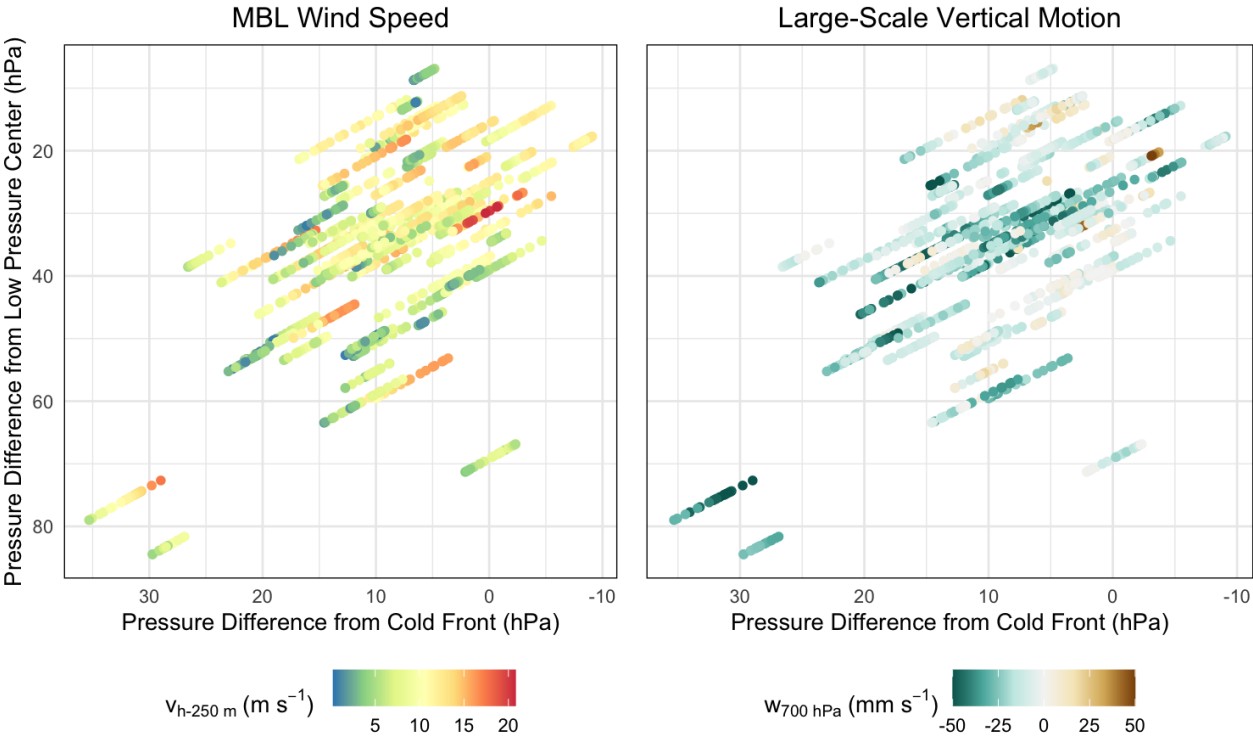

**Figure 2.** For all ACTIVATE tandem flights during postfrontal conditions, we measure the smallest pressure difference to cold front (x-axis) and low-pressure center (y-axis). Left and right panels show the same data points but different shadings: 250 m horizontal wind speed (left) and large-scale vertical motion at 700 hPa (right), where 700 hPa is expected to represent free tropospheric conditions.

surface fluxes are greater (median of 550 W m$^{-2}$ compared to 300 and 190 W m$^{-2}$, respectively), where latter is expected to
increase with near-surface wind speed and surface-air differential in temperature and humidity. The above example season is representative of ACTIVATE's earlier off-summer deployments in spring 2020 and fall 2020 through spring 2021, which are shown in Fig. A2. Note that sample size is smaller in Fig. A2, preventing box-whisker plots from reliably producing whiskers.

### 3.3 Potential for Case Studies

To utilize the ACTIVATE field observations for Lagrangian LES and SCM simulations, flights ideally must have captured:

1. eastward flow associated with typical postfrontal synoptic conditions free of high-level clouds, for example seen via the trajectories in Fig. 1,

2. relatively large MCAO indices associated with well-defined cold-air outbreak conditions,

3. MBL flow along near-parallel Lagrangian trajectories to effectively capture the downwind evolution of the boundary layer flow, and

4. sufficient fetch to sample MBL evolution (i.e., suffcent distance to detect process signatures, such as CCN decrease from precipitation formation) and with key instruments operating (e.g., RSP being the most volatile one owing to limited sun-object-observer geometries for operation), in the best scenario probing an air mass more than once to establish a true Lagrangian sample (i.e., free of stationarity assumptions)

Additionally, we indicate whether flights capture:

5. both warm and cold precipitation particles, which are expected to affect cloud regime transition when appearing in substantial concentrations,

6. recent new particle formation that undergoes particle growth and could result in aerosol particles large enough to act as cloud condensation nuclei (Zheng et al., 2021; Tornow and colleagues, in prep.), in particular during enhanced updraft speeds where more particles are activated (e.g., Kirschler et al., 2022).

These criteria are defined in greater detail in Sec. 2.5. Fig. 4 (left) presents a score sheet, verifying criteria for all postfrontal flights. All flights fulfill at least three scores, for example 2 March 2020 (RF015) only providing eastward flow and warm precipitation-type hydrometeors while 26 January 2022 leg 1 (RF111) only provided all three precipitation-type hydrometeors (i.e., drizzle, rain, and frozen hydrometeors) and good alignment between flight and trajectory, leaving all other criteria unchecked. About two thirds of the flights present an eastward flow, about half have high MCAO indices, and about half are

free of high clouds. Four out of five flights attributed a large portion (i.e., more than 20 %) of their resources aligned with a Lagrangian trajectory, but only two out of five captured a large enough fetch (i.e, more than 200 km) and about half had RSP data available for much of the downwind portion (i.e., more than 30 %). While air mass revisits are nearly unavoidable on mostly non-linear flight tracks, two out of five flights did so with horizontal legs long enough (i.e., at least one minute) to gather measurements from all instruments or even obtain a statistic based on a small amount of data. In about two thirds of all

flights the revisit happened after a significant amount of time (i.e., at least 1.5 hours), and one in five flights had RSP observations available when revisiting. All flights captured warm-phase precipitation hydrometeors, but only about two thirds of all flights collected frozen hydrometeors, in many instances aligning with an unchecked MCAO index criterion. About half the flights show signs of new particle formation and about a third measured elevated aerosol concentrations during non-negligible portions of the flight (i.e., more than 5 % of a flight saw concentrations greater than 10,000 cm$^{-3}$).

Selected flights (marked through a red outline in the score sheet) show generally a greater score. For example, the pair of flights on 29 March 2022 (RF147 and RF148) fulfills all required criteria, rendering it a particularly strong case for Lagrangian modeling and analysis. The pair of flights on 1 March 2020 (RF013 and RF014) fulfills nearly all criteria, only lacking RSP along quasi-Lagrangian stretches during the first flight and elevated MCAO indices during the second one. The combination of both flights is still expected to produce excellent case data, by, for example, relying on upwind data from the first flight and

downwind data from the second one. The single flight on 3 February 2021 (RF044) checks all but one required criterion that should still produce excellent case data: the relatively short cloud regime transition was captured by the flight (Fig. A3), but resulted in a quasi-Lagrangian stretch shorter than 200 km. The single flight on 29 January 2021 (RF042) lacks two Lagrangian aspects, including long enough horizontal legs and RSP being available. The flights on 13 March 2022 (RF137 and RF138)

lack RSP during Lagrangian airmass revisits and the first flight covers a quasi-Lagrangian stretch shorter than 200 km. The pair of flights on 11 January 2022 (RF100 and RF101) satisfy most criteria during the first flight, with the expectation of prolonged Lagrangian revisits and with RSP being available, while the second flight lacks all quasi-Lagrangian aspects. The pair of flights on 18 January 2022 (RF105 and RF106) only lacks RSP availability during the quasi-Lagrangian stretch of the second flight, and Lagrangian sampling aspects in both flights. All selected flights show both liquid and frozen hydrometeor particles being present. Only a few flights show elevated aerosol particle counts, for example, the second flight on 1 March 2020 (RF014), the single flight on 29 January 2021 (RF042), and the second flight on 29 March 2022 (RF148) while indication of new particle formation is present in most flights with the exception of 3 February 2021 (RF044) and 11 January 2022 (RF100 and RF101).

Complementary to the score sheet, we summarize which portion of the greater cloud lifecycle was probed. Ideally, flights captured both cloud formation as well as the cloud regime transition to a broken state farther downwind. Fig. 1 provides an example where the aircraft first passed the cloud-free area off the eastern seaboard, then the formed cloud deck including its brightest location, and lastly accessed the dimmer, broken cloud field farthest east before turning around.

To quantify this exposure to different stages, we perform the following steps. Per Lagrangian trajectories (launched every 10 min along each flight track) we collocate GOES16-based cloud cover (Sec. 2.3), available every 20 minutes. From the resulting timeline of cloud cover, we then extract two events:

1. cloud formation, defined as the first instance exceeding a cloud cover of 75 %, and

2. cloud breakup, defined as the first instance of cloud cover below 75 % after formation.

Fig. A4 provides an example for all trajectories during the second flight on 29 March 2022 (RF148). Per cloud lifecycle event we then calculate its timing and distance to the flight (i.e., the trajectory launch).

Fig. 4 (right) shows the distribution of distances by using all trajectories. Negative values (also highlighted by gray shading as well as "ACTIVATE's reach") mean that ACTIVATE aircraft were located downwind of an event (e.g., in clouds that increased beyond 75 % past cloud formation), likely passing the event on its way out, whereas positive values (white shading) mean that an event was located farther downwind than seen by aircraft (e.g., in clouds that have yet to increase towards 75 % cloud cover for cloud formation). The above ideally translates into both events – cloud formation (red) and cloud breakup (cyan) - being inside the negative range. Most flights only probed the formation of clouds (i.e., red values are in the negative range) and left their regime transition unobserved (i.e., cyan values are in the positive range). For example, Fig. 1's flight on 29 March 2022 (RF148) just reached the breakup stage, similar to the other selected flights. Where trajectories intercepted a cloud deck that was too thin or too broken (to exceed a cloud fraction of 75 %), such as 22 September 2020 (RF037), no red or cyan curves are shown.

In a few instances flights probed both stages robustly. Seen in combination with the score sheet, these flights, however, lack other qualities. For example, on 30 November 2021 (RF094) and 10 December 2021 (RF099) RSP was unavailable and there was no Lagrangian element. In general, few flights are of high score and also show proximity to both lifecycle events, such as the second flight on 28 February 2020 (RF011), which however displayed an atypical cloud cover evolution, likely due to uncommon upward motion (Li et al., 2021) in connection with an apparent smaller front (Fig. A3).

### 3.4 Cloud Properties during flights

Next, we survey cloud properties seen by ACTIVATE that enable us, for example, to develop an expectation for prevalent
microphysical processes (e.g., mixed-phase processes like riming), as discussed in Section 4. We collect cloud macrophysical
retrievals from HSRL-2 remote sensing (cloud-top height and temperature), and microphysical properties measured by FCDP
and 2DS in-situ probes (cloud droplet number and frozen hydrometeor concentrations) as explained in greater detail in Sec. 2.4.

Fig. 5 summarizes each property through three percentiles, $5^{th}$, $50^{th}$ and $95^{th}$ per flight. Across all cases clouds generally
occupy the lower 2.5 km of the atmosphere, in a few instances extending to 3.5 km (e.g., 30 November 2021, RF094), typically
translating into cloud-top temperatures between -10 and +5 °C with a few instances of smaller and greater temperatures.
All cases span a wide range of cloud droplet number concentrations, with median values between 100 and 500 cm$^{-3}$, while
extremes (here $95^{th}$ percentiles) can exceed values of 1500 cm$^{-3}$ in rare cases, for example, on 24 January 2022 (RF109
and RF110) as well as 8 and 9 March 2021 (RF051 and RF052, respectively). Most frozen hydrometeor concentrations show
median values around 1 L$^{-1}$, but extremes can reach concentrations that are two orders of magnitudes higher, such as on 28
February 2020 (RF010 and RF011).

The selected subset (shown in red) shows relatively typical cloud-top heights (median values ranging between 1500 and 2100
m), but colder cloud-top temperatures (median values between -12 to -5 °C). Droplet number concentrations show relatively
typical median values (between 200 and 500 cm$^{-3}$), but can extend beyond 1000 cm$^{-3}$ in their extremes (e.g., 1 March 2020,
RF013 and RF014, and 29 March 2022, RF147 and RF148). The selected cases show frozen hydrometeor concentrations within
the overall range.

### 3.5 Evolution of Selected Cases

Lastly, we examine the cloud macro-physical as well as meteorological evolution of the selected cases. Drawing from GOES-
16 cloud cover along trajectories (e.g., seen in Fig. A5), we compute time with respect to their surface flux maximum. As
shown in Fig. 6a, we find a uniform cloud cover increase across all cases, with overcast conditions reached around 0 h for all
cases, except 1 March 2020. Thereafter, cloud cover decreases as part of the cloud regime transition and reaches a broad range
of levels, ranging anywhere between 30 and 80 %, bracketed by 1 March 2020 and 13 March 2022 on the low and high end,
respectively. The selected days also display a diverse evolution in cloud-top temperature (Fig. 6b), roughly matching the range
probed by ACTIVATE remote sensing (Fig. 5) near 0 h. MERRA2-based MCAO indices along trajectories (Fig. 6c) show a
collective decrease after the maximum, with cases retaining their relative strength (e.g., 11 January 2022 remains strongest
throughout). MERRA2-based large-scale subsidence shows vastly different values with strong fluctuations over time, but al-
ways displaying negative median values (i.e., downward motion, Fig. 6d).

In summary, we find ACTIVATE to have successfully sampled postfrontal conditions, probing a broad range of locations
within postfrontal sectors. These conditions, naturally, deviate from synoptic conditions across the overall season. A few cold-
air outbreak flights that tend to deviate more strongly in their meteorology show qualities that are desirable for Lagrangian LES

and SCM cases: they cover a sizeable portion of a typical boundary flow, capturing cloud formation and at least brief portions of the cloud regime transition to the broken state via key remote sensing and in-situ instruments. Cloud properties show sizable differences across those flights, with selected flights typically at the lower end of cloud-top temperature but all other properties within the pack.

## 4   Discussion

The cloud properties seen in Fig. 5 lend themselves to speculate about dominant processes. For example, a great range in cloud-top height could result from substantial MBL deepening with fetch. Deepening against FT subsidence translates into substantial entrainment at the MBL top, mixing in FT air that typically shows a lower concentration of aerosol available as
CCN and thereby dilutes MBL concentrations  (Tornow et al., 2022). As an example among the selected flights, 1 March 2020 shows a spread of about 1000 m in cloud-top height and was assessed to have a peak entrainment rate of 12 cm s$^{-1}$. Other selected flights also largely line up with the boundary flow and cover similar ranges in cloud-top height, except for 11 January 2022 showing about 1500 m. Second, the presence of large frozen hydrometeor concentrations and the prevalent range in cloud-top temperatures imply secondary ice processes at play. 28 February 2020 shows up to $N_i <= 100$ L$^{-1}$ reaching,
like many other flights, cloud-top temperatures of -13 to -4 °C (i.e., 5$^{\text{th}}$ and 95$^{\text{th}}$ percentiles) that may be favorable to ice multiplication processes that are considered highly uncertain (e.g., Fridlind and Ackerman, 2018; Korolev et al., 2020; Korolev and Leisner, 2020; Seidel et al., 2024). Frozen hydrometeors have often shown signs of riming during ACTIVATE (Seethala et al., 2024), which may in turn impact the cloud regime transition (Tornow et al., 2021).

The selected cases constitute generally strong MCAOs and should provide excellent targets for weather and climate model
development. While ACTIVATE covers a wider spectrum of conditions, the selected subset forms a robust statistical sample of unique mixed-phase clouds. The range in meteorological forcing and resulting cloud cover and cloud-top temperature evolution provides a testbed for model developers to explore the inclusion of uncertain mixed-phase processes and benchmark against a comphrehensive set of observational targets from ACTIVATE and satellite. The wide range in cloud-top temperature has the potential to serve as proxy of a warming climate and assess cloud-climate feedback.
The criteria utilized in Section 3 may provide a blueprint for future deployments. Building on the growing experience of setting up Lagrangian case studies for LES and SCM in the community, the list of criteria would likely be refined or extended. For ACTIVATE, the airbase at LaRC was conveniently located upwind from postfrontal clouds, thereby allowing to automatically sample initial upwind conditions needed for case studies, while the more challenging part was to reach downwind locations. Therefore, many criteria target the latter challenge. For aircraft campaigns that are located at the downwind portion
additional criteria may be required, for example, the ability to gather upwind conditions.

Where flights fall short of capturing cloud regime transitions, the use of satellite data can be helpful. As done in pre-campaign (Tornow et al., 2023) and campaign efforts (Tornow and colleagues, in prep.), low-earth orbiting as well as geostationary satellite retrievals may provide observational constraints. Among the most valuable constraints are instantaneous total

liquid water path retrievals from microwave radiometers aboard a fleet of low-earth orbiting satellites (Elsaesser et al., 2017),
complementing aircraft retrievals of liquid water path where available (e.g., Ephraim et al., 2024), and cloud cover, obtained
from day-night COD retrievals using geostationary imagery (used here to identify lifecycle stages).

## 5 Conclusions

Our analysis of ACTIVATE's tandem flights during postfrontal conditions supports the following conclusions:

1. Aircraft data covered a wide range of locations within postfrontal sectors and represent typical MBL horizontal and FT
vertical wind speeds.

2. The dedication to postfrontal clouds facilitated distinct meteorological conditions that exceed the seasonally typical con-
ditions of FT subsidence, MCAO indices, and resulting turbulent surface fluxes. A proposed subset of cold-air outbreak
flights that are well suited to Lagrangian analyses constitutes even greater MBL wind speed, FT subsidence, MCAO in-
dices, and surface fluxes, compared to all postfrontal flights. The selected flight days include 1 March 2020, 29 January
2021, 3 February 2021, 11 and 18 January 2022, and 13 and 29 March 2022.

3. Criteria that aim to measure the ability of flight data to serve in Lagrangian LES and SCM case studies reveal a wide range
of qualities. The selected flights typically cover a large fetch in MBL flow direction, have key instruments operating,
and often exhibit evidence of specific aerosol and cloud processes. These flights experience the formation of clouds, but
have typically only briefly visited the cloud regime transition toward a broken cloud deck. The use of satellite retrievals
offers a way to obtain observational constraints farther downwind.

4. Remote sensing and in-situ probes reveal a wide range of cloud macro- and microphysical properties that suggest the
presence of dominant processes, such as strong FT entrainment, riming, and secondary ice formation.

ACTIVATE data provide a unique resource to study cloud controlling processes in MCAOs and improve the representation
of clouds and aerosol in upcoming ESM improvement work.

*Code and data availability.* MERRA-2 fields were downloaded from https://www.earthdata.nasa.gov/. GOES-16 imagery was downloaded
from the Space Science and Engineering Center (SSEC), University of Wisconsin–Madison, using McIDAS version 4. GOES-16 re-
trievals (NASA/LaRC/SD/ASDC, 2021a) and ACTIVATE flight measurements (NASA/LaRC/SD/ASDC, 2023, 2021b) are publicly avail-
able via data archive at ACTIVATE's field data repository: https://asdc.larc.nasa.gov/soot/search. The code to detect frontal regions using
Hewson spatial gradients, is provided at https://github.com/coecms/frontdetection/tree/main and was translated into R language. Code that
scores flights and summarizes cloud properties is available upon request.

# Appendix A: Supporting Figures

*Author contributions.* FT carried out the analysis and prepared the manuscript. AF, AS, AA, GT, PZ, CV, SK and BC provided feedback on the analysis and manuscript. SC and PZ provided identification of postfrontal flights. DP provided GOES-16 retrievals. CV and SK provided in-situ cloud data.

*Competing interests.* At least one of the (co-)authors is a member of the editorial board of Atmospheric Chemistry and Physics.

*Acknowledgements.* This work was supported by ACTIVATE, a NASA Earth Venture Suborbital-3 (EVS-3) investigation funded by NASA's Earth Science Division and managed through the Earth System Science Pathfinder Program Office Division (Grant No. 80NSSC19K044). The authors thank the ACTIVATE team for helpful discussion and for support with measurements and quality control. AF was supported by the NASA Modeling, Analysis and Prediction Program funding for ModelE development. AS was supported by NASA grant

no. 80NSSC19K0442. PZ and SC gratefully acknowledge funding support from NASA grant 80NSSC19K0390. CV acknowledges support from the Deutsche Forschungsgemeinschaft (DFG, German Research Foundation) under project ID 428312742 (TRR301) and ID 522359172 (SPP HALO). SK was supported by the European Union's Horizon Europe program within the Single European Sky ATM Research 3 Joint Undertaking through the CONCERTO (grant no 101114785) and CICONIA (grant no 101114613) projects.

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

**Table 1.** Criteria to indicate specific qualities of each postfrontal flight and its synoptic condition represented through Lagrangian trajectories.

| Statement | Data and Methodology |
|---|---|
| *Primarily Eastward flow* | Uses Lagrangian trajectories and checks whether most meridional wind components are positive within the first 12 hours |
| *Maximum MCAO index > 10 K* | Uses Lagrangian trajectories and computes the median across all trajectories' maximum MCAO = $\theta_{\mathrm{srf}} - \theta_{850\,\mathrm{hPa}}$ |
| *High cloud fraction < 10 %* | Uses Lagrangian trajectories and checks whether collocated GOES-16 cloud properties contain show fewer than 10 % of data points with a cloud-top height above 5 km. |
| *Downwind portion with respect to flight > 20%* | Uses flight track coordinates and Lagrangian trajectories to first determine the portion of the track falling within 20 km of each trajectory and then report whether any portion is greater than 20% |
| *Downwind distance > 200 km* | For extent of alignment (see above, aligning more than 20% of the track with a trajectory), checks whether the downwind distance along the flight track exceeds 200 km |
| *Downwind RSP availability > 30%* | For long downwind distances (see above, a distance greater than 200 km for cases of great alignment), checks whether RSP is available for more than 30% of the time |
| *Minimum duration of Lagrangian legs > 1 minute* | From all portions of the flight track falling within 20 km and 10 minutes of a trajectory, quantifies the number of samples taken at up- and downwind portions, and reports whether the minimum duration covers at least one minute |
| *Lagrangian span > 1.5 hours* | From all portions of the flight track falling within 20 km and 10 minutes of a trajectory, measures the greatest time span across samples |
| *Lagrangian RSP availability* | From all portions of the King Air track, measured whether RSP is available during at least 100 acquisitions (i.e., totaling about 1.5 minutes of data) at up- and downwind portions |
| *Frozen hydrometeors measured* | Checks whether 2DS reported any particles classified as frozen hydrometeors |
| *Drizzle-sized particles measured* | From FCDP and 2DS size distributions of liquid particles, determines whether the count of particles with diameter greater than 40 micrometers is above zero (note that particles smaller than 108 nm are all assumed liquid, Sec. 2.4) |
| *Rain-size particles measured* | From 2DS size distributions of liquid particles, determines whether the count of liquid particles with diameter greater than 100 micrometers is above zero |
| *Recent particle formation frequency > 5 %* | Determines the portion of the flight track that shows a $CN_{3\,\mathrm{nm}}/CN_{10\,\mathrm{nm}}$ ratio greater 1.8 (e.g., Corral et al., 2022; Namdari et al., 2024) and reports whether the threshold is exceeded more than 5 % of the time |
| *High aerosol concentration frequency > 5 %* | Determines the portion of the flight track that shows a $CN_{10\mathrm{nm}}$ greater than 10.000 $\mathrm{cm}^{-3}$ and reports whether the threshold is exceeded more than 5 % of the time |

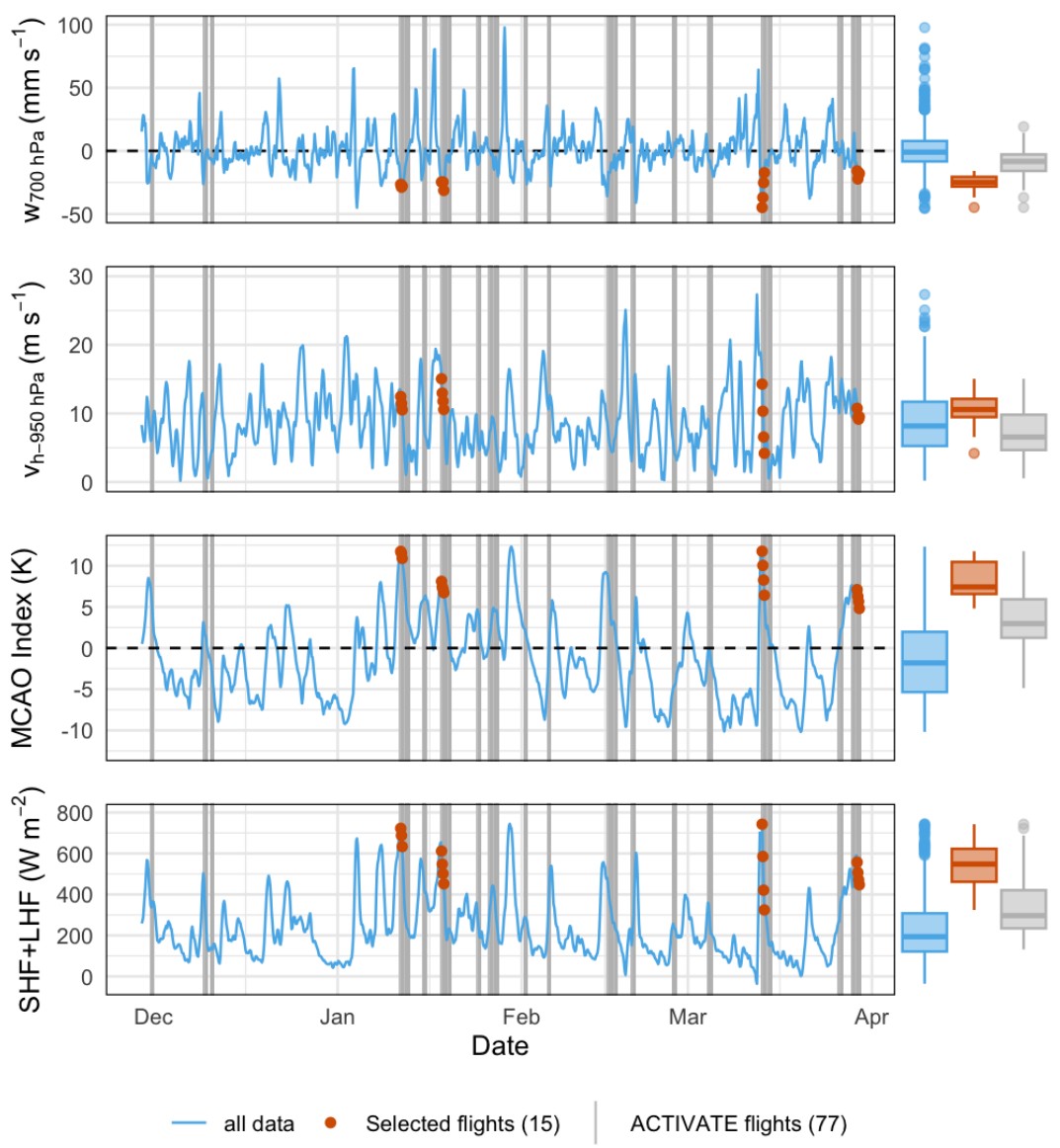

**Figure 3.** Averaged over the ACTIVATE domain and during the period of 29 November 2021 until 30 March 2022, we extract 3-hourly meteorological parameters from MERRA-2 (from top to bottom): (1) large-scale vertical motion at 700 hPa, (2) horizontal wind speed at 950 hPa, (3) MCAO index, defined as MCAO $= \theta_{\mathrm{srf}} - \theta_{850\,\mathrm{hPa}}$, and (4) lumped turbulent surface fluxes. Compared to the complete time series (blue lines), ACTIVATE flights (gray vertical bars) targeted specific conditions and selected flights (red dots) form a particular subset. Box-whisker plots (right) mark the overall distribution of all three groups. The legend also lists the number of data points used for box-whisker plots.

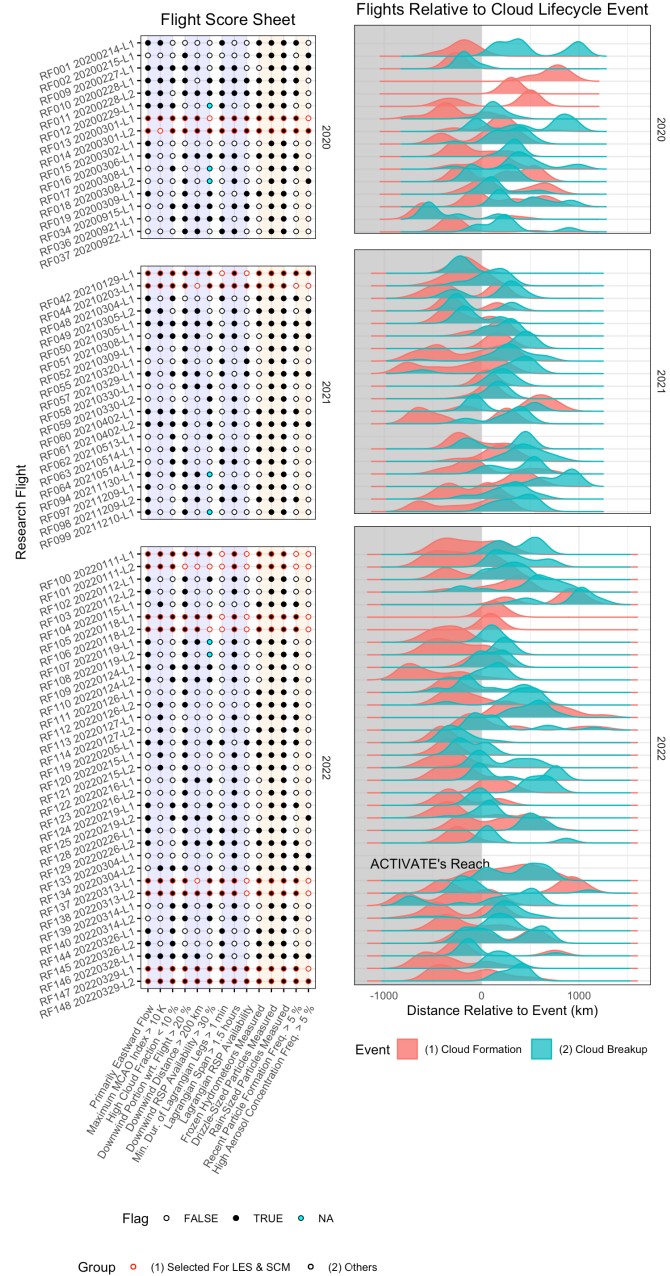

**Figure 4.** For all tandem flights under postfrontal conditions, we summarize the veracity of criteria in a score sheet (left, explained in greater detail in Sec. 2.5, grouped into categories marked by shaded areas with blue and orange marking criteria of primary and secondary importance, respectively) and display the relative distance of flight data compared to cloud formation and breakup (right, gray areas mark fetch values reached by ACTIVATE, explained in greater detail in Sec. 3.3). L values next to dates (format: YYYYMMDD) correspond to launch number (e.g., L1 is the first flight [i.e., launch] of that given day).

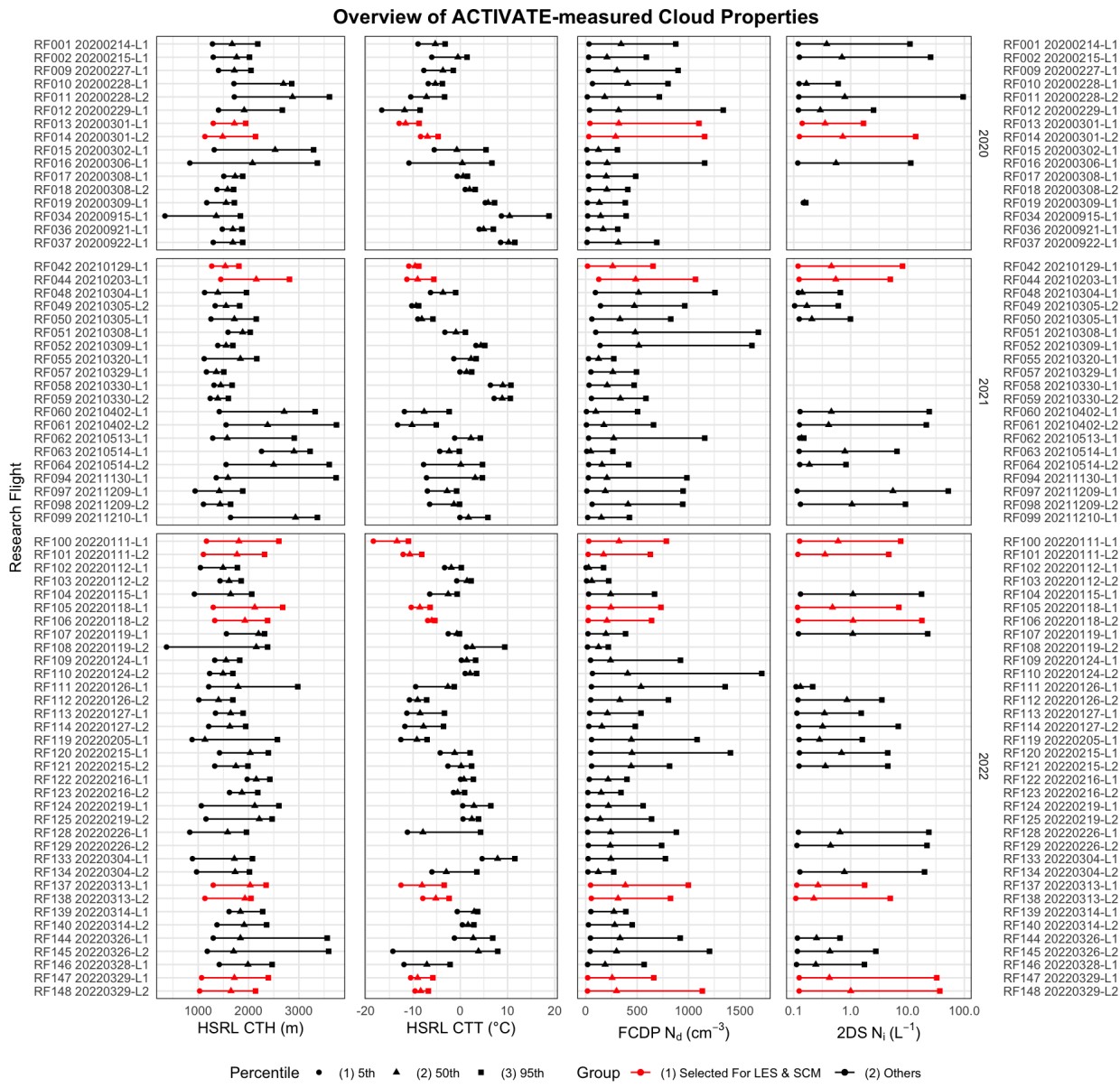

**Figure 5.** Using each flight's in-situ and remote sensing data, we summarize various cloud properties (from left to right): cloud-top height (CTH), cloud-top temperature (CTT), cloud droplet number concentration, $N_d$, and frozen hydrometeor number concentration, $N_i$. Percentiles (symbols) indicate the distribution, with horizontal lines spanning $5^{th}$ and $95^{th}$ percentiles. Note that the x-axis of the rightmost panel is shown in logarithmic scale. Selected cases are shown in red.

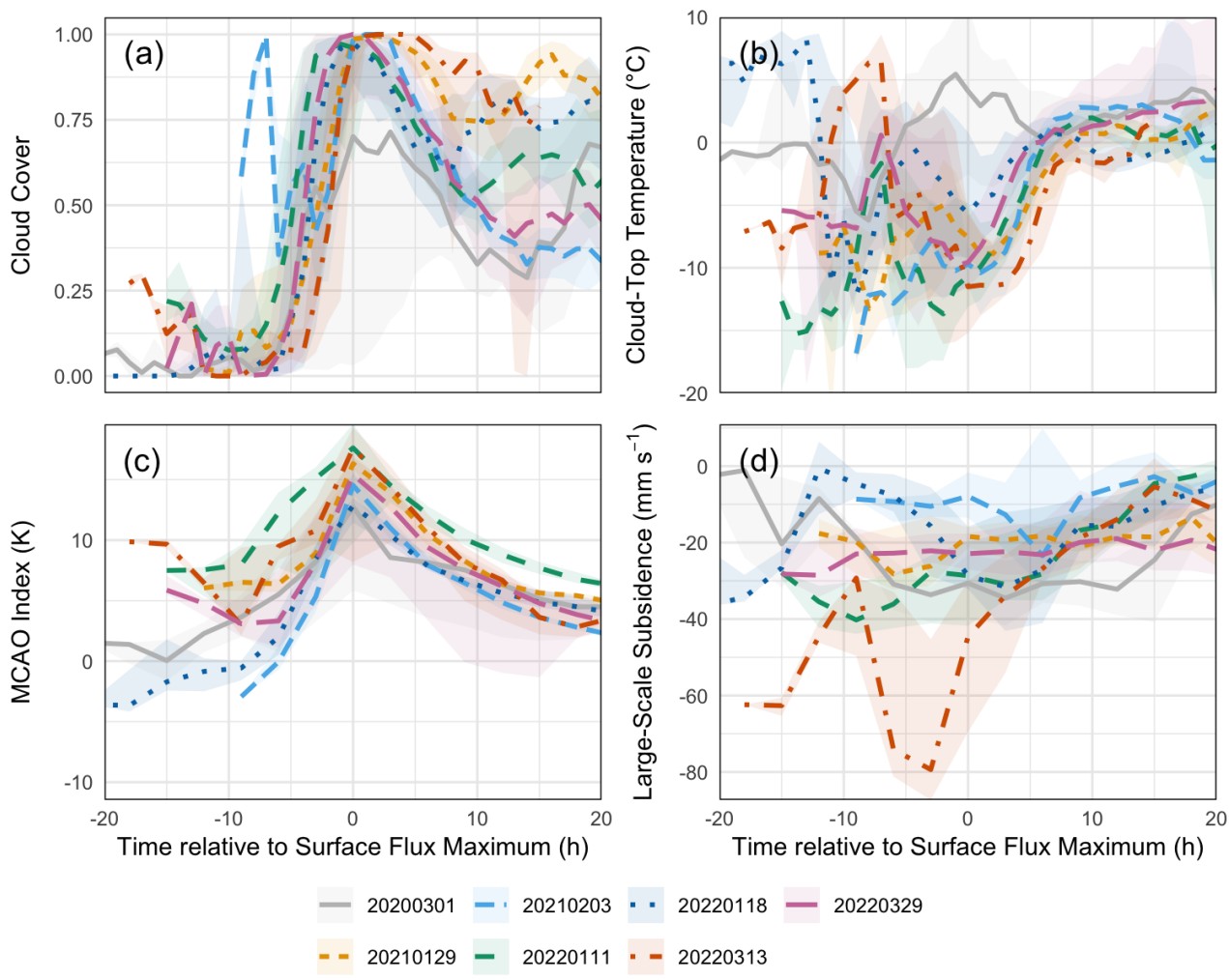

**Figure 6.** Relative to each trajectory's time of surface flux maximum, we present statistics (i.e., median shown as lines and interquartile range shown as shading) per selected case, showing GOES-16 cloud properties and MERRA-2 meteorological boundary conditions: (a) cloud cover, (b) cloud-top temperature, (c) MCAO index, and (d) $w_{700\,\text{hPa}}$.

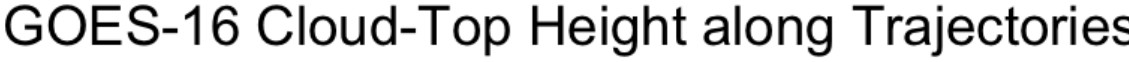

**Figure A1.** From GOES-16 cloud-top heights that we collocated along all trajectories, we produce stacked density functions for two categories

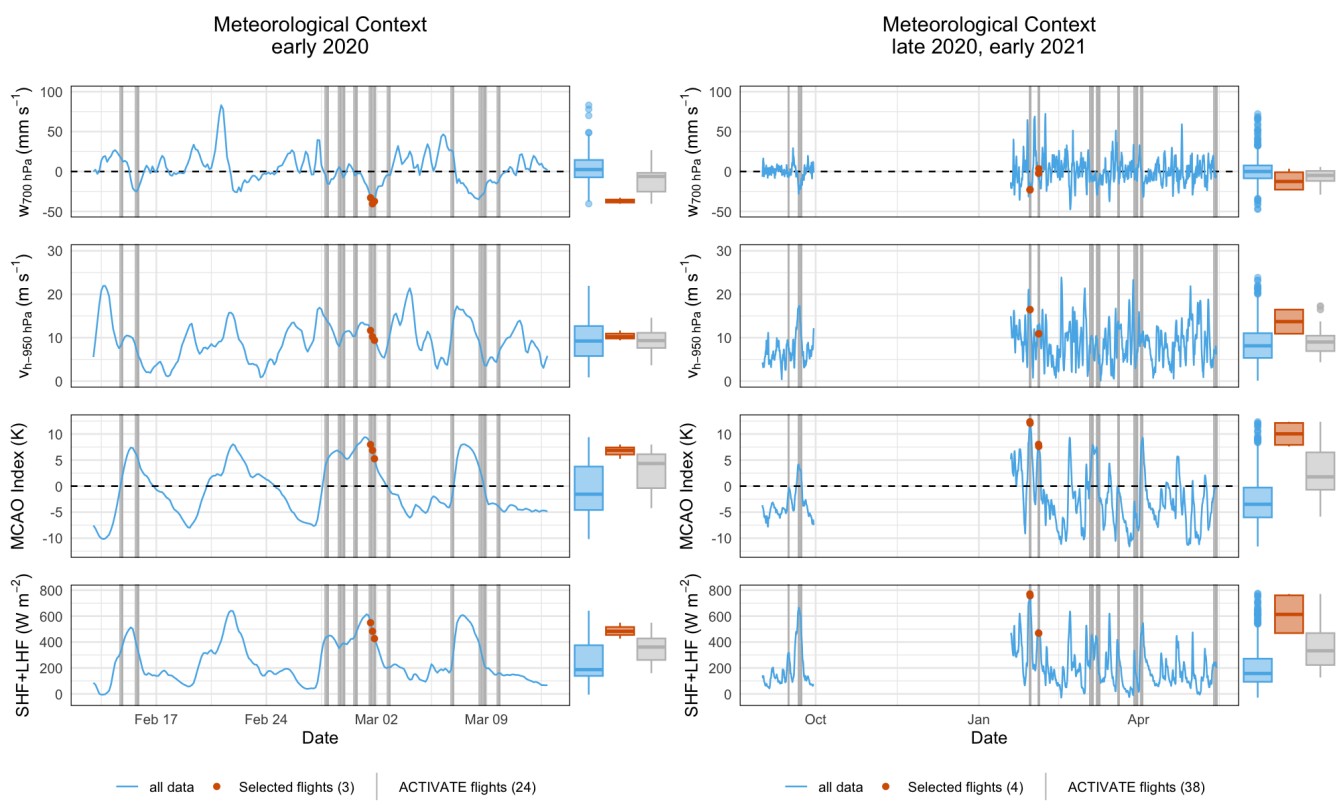

**Figure A2.** Like Fig. 3, here shown for the two earlier deployments. Note that both panels contains too few data points in *Selected Flights* to produce whiskers.

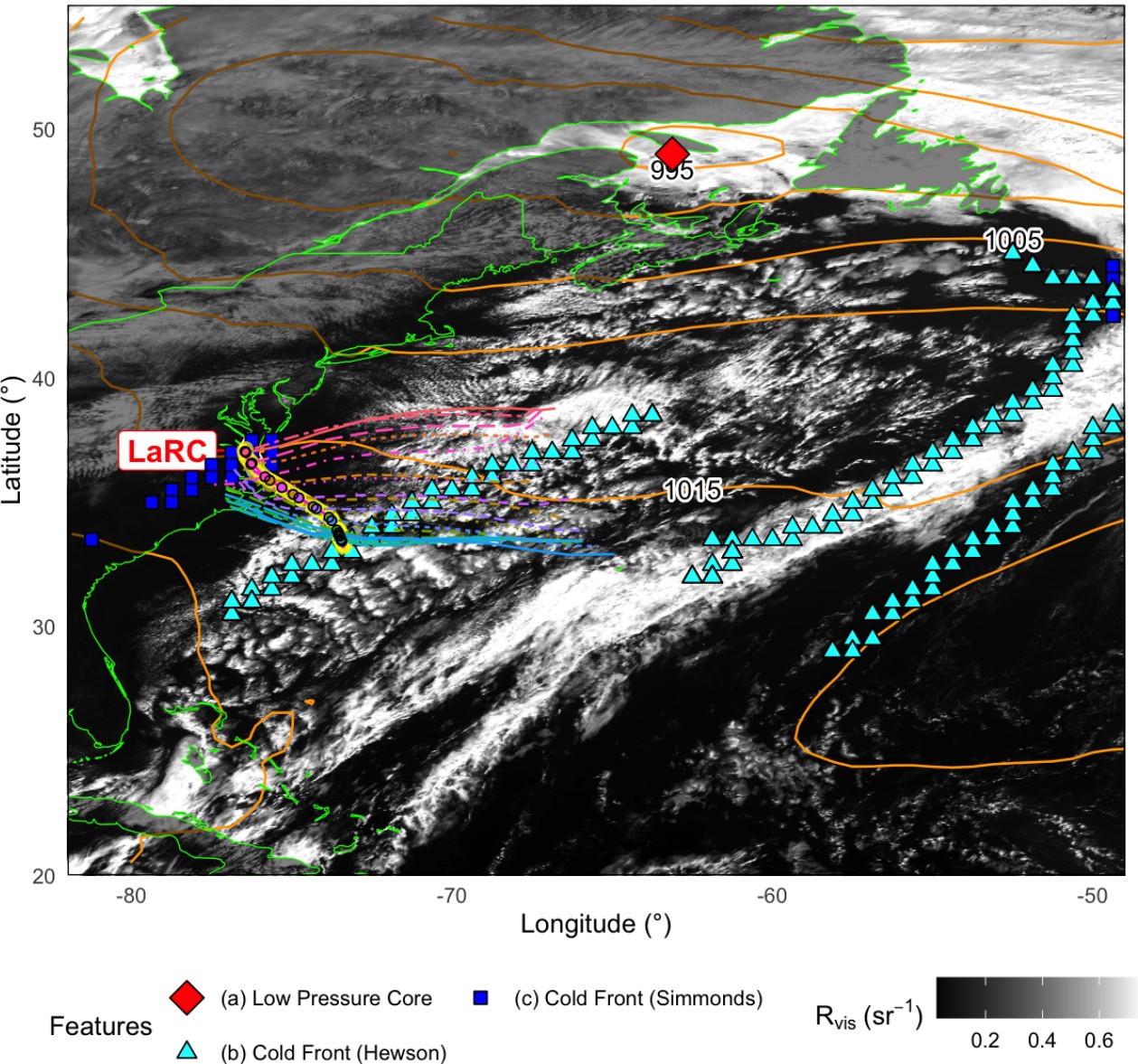

**Figure A3.** Similar to Fig. 1, we present data for 28 February 2020.

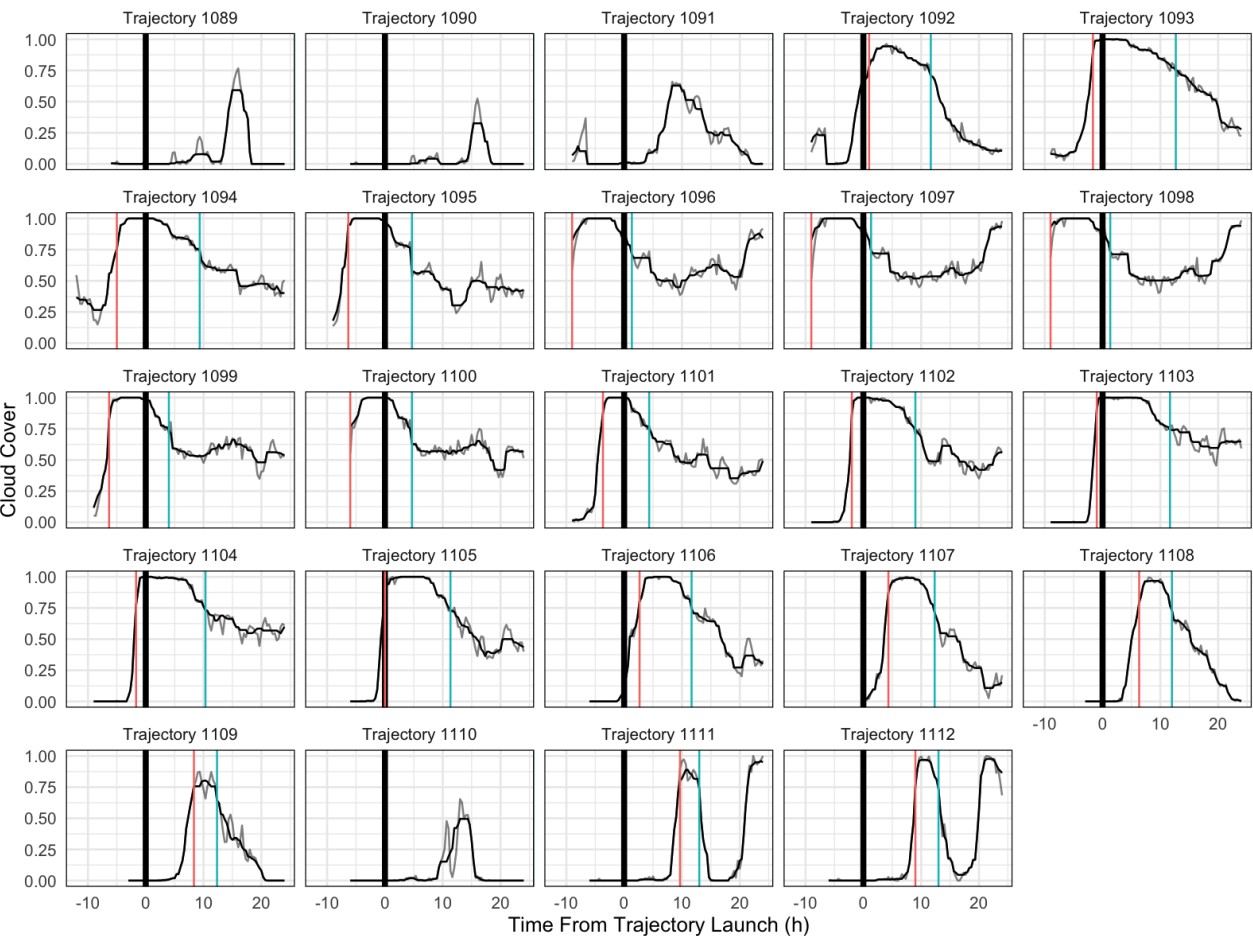

**Figure A4.** Timelines of cloud cover (thin gray line) along each trajectory (panel by panel) produced for the second flight on 29 March 2022 (shown in Fig. 1). A low-pass filter smoothes timelines (thinner black line). The relative position to flight data (thick, black, vertical line) to cloud formation (red) and cloud breakup (light blue) is measured as the difference in time and downwind distance.

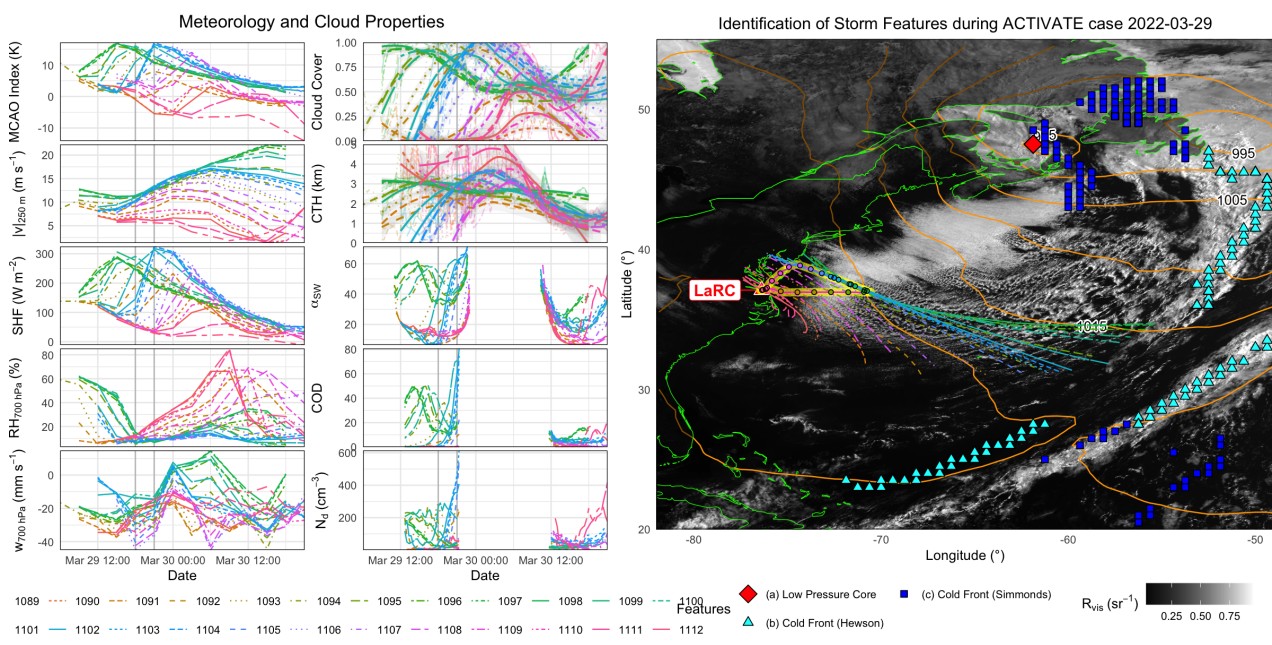

**Figure A5.** Various MERRA-2 meteorological and GOES-16 cloud properties (left) along all trajectories that are shown on the map (right), which is identical to Fig. 1.

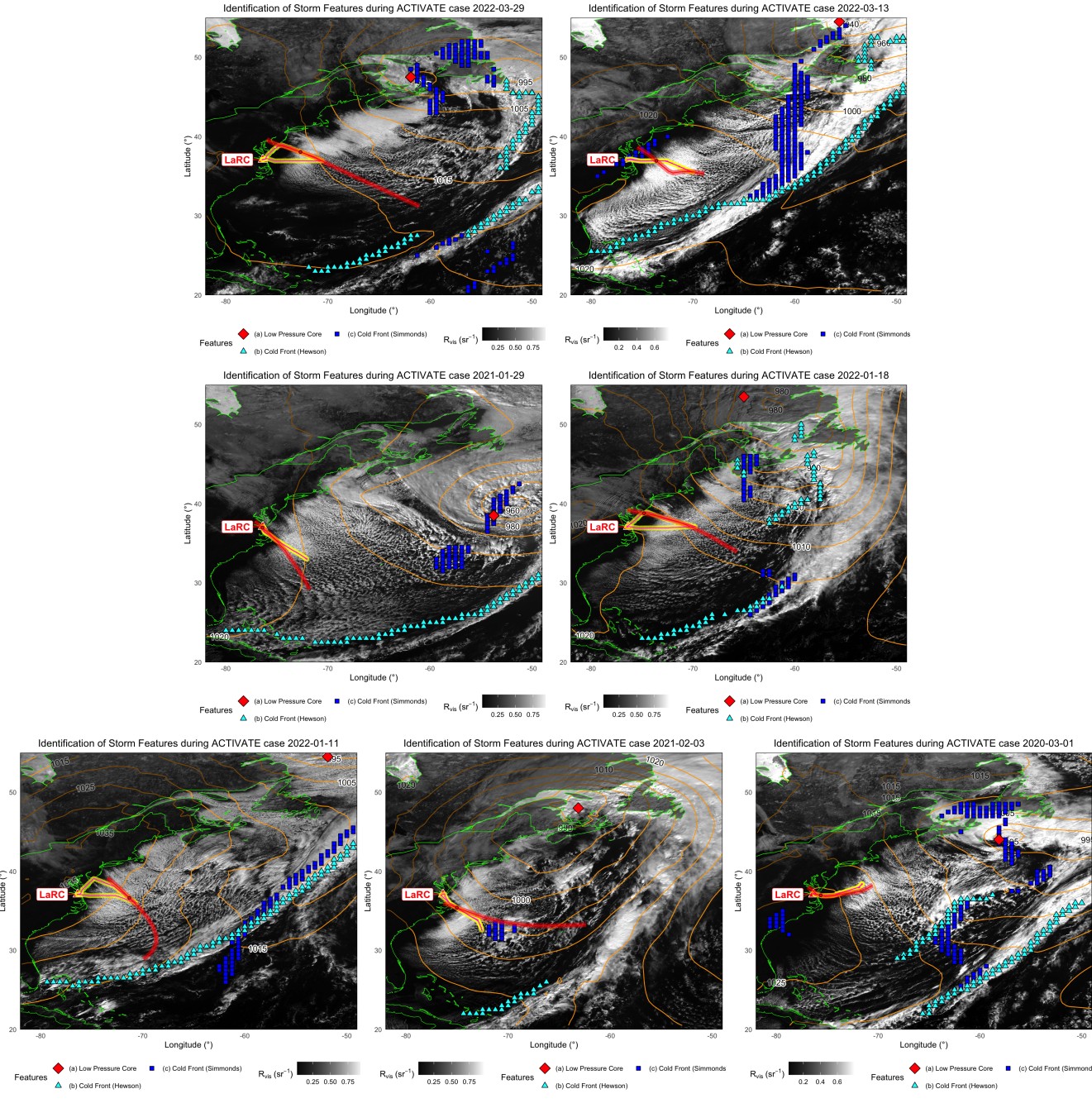

**Figure A6.** Similar to Fig. 1, but here shown for the first flight of each selected day (yellow) with the trajectory (red) that maximally connects both flights, determined through the maximum fraction of timestamps within 20 km and 1 hour across trajectory.