# Peer review of "Measurement Report: A survey of meteorological and cloud properties during ACTIVATE's postfrontal flights and their suitability for Lagrangian case studies"

_EGUsphere, 2024_

## Author Comment (AC1)

We thank both Reviewers for their valuable input. Please find concerns addressed point-by-point. A response to each concern (black, bold) is shown in blue, while the resulting change to the manuscript is shown in blue, italic font. Where line numbers are listed in our response, we refer to the revised version containing tracked changes.

**Reviewer 1**

This measurement report is about marine cold-air outbreak flights during the ACTIVATE campaign. The authors report on the synoptic and meteorological conditions during flights, cloud properties, and choose specific flights of interest for Lagrangian modelling studies.

Overall, I think that this is a nice measurement report that is certainly helpful for anyone interested in choosing specific cases for a modelling study. My main concern is that that the selected cases all appear rather similar. Below I made some comments that the authors should consider before publication.

**General Comments**

1. **My largest concern is that it appears that all the 'selected' flights are fairly similar in terms of their characteristics, yet some potentially interesting cases such as 29 Jan 2021 and 28 Feb 2020 are not selected. Given that some of the selected cases have already been studied in published work, I think the report could benefit from some additional discussion about what makes each selected case unique and what would be interesting to study about them. At the moment it seems that no matter which of the selected cases is chosen one would get similar results while other cases with more different characteristics are left out. For instance, what would be the difference between choosing the 1 Mar 2020 and the 13 Mar 2022 flights, they seem very similar. Intentionally or not the authors might be steering the research to specific types of cases that then might end up being overrepresented in the literature, while other relevant cases are not studied.**

The Reviewer pointed out two days that were left out in our original submission. We decided to add '20210129' but leave out '20200228'. Latter case shows an atypical cloud cover evolution that could be connected to uncommon upward motion (Li et al., 2022). Former case is already included in the newly added figures below and complements the spectrum through its elevated cloud cover after breakup. We added the following text to Section 3.3 and a figure to the appendix displaying '20200228' in its synoptic framework:

*ll 293-294: The single flight on 29 January 2021 (RF042) lacks two Lagrangian aspects, including long enough horizontal legs and RSP being available.*

*and*

*ll 327-328: ..., such as the second flight on 28 February 2020, which however displayed an atypical cloud cover evolution, likely due to uncommon upward motion (Li et al., 2022) in connection with an apparent smaller front (Fig. A3).*

[Figure]

**Fig. A3:** *Similar to Fig, 1, we present data for 28 February 2020.*

In response to the Reviewer's concern, we performed additional analysis to highlight the differences across the selected flights. From GOES16-based cloud properties and MERRA-2 based meteorological properties along trajectories, we computed median and interquartile range as a function of time from the surface flux maximum. When superimposed, the various days show a consistent cloud formation (i.e., an increase in cloud cover), but a vast range of cloud breakups (i.e., decrease in cloud cover), reaching cloud cover levels anywhere between 30 and 80 % a few hours after the cloud cover maximum. The two selected flights that the Reviewer pointed out, including '20200301' and '20220313', bracket the bunch of days. In addition, all selected days have a unique evolution in cloud-top temperature, which – as we already discuss in Section 4 – may enable mixed-phase cloud-aerosol processes, as well as meteorological boundary conditions (here shown for MCAO index and large-scale subsidence). All in all, the set of selected days does provide ample variety while generally falling into the category of

strong marine cold-air outbreaks. We added the following subsection at lines 346-359 and Figure 6:

**Section 3.5: Evolution of Selected Cases**

*Lastly, we examine the cloud macro-physical as well as meteorological evolution of the selected cases. Drawing from GOES-16 cloud cover along trajectories (e.g., as seen in Fig. A5), we compute time with respect to their surface flux maximum. As shown in Fig. 6a, we find a uniform cloud cover increase across all cases, with overcast conditions reached around 0 h for all cases, except 1 March 2020. Thereafter, cloud cover decreases as part of the cloud regime transition and reaches a broad range of levels, ranging anywhere between 30 and 80 %, bracketed by 1 March 2020 and 13 March 2022 on the low and high end, respectively. The selected days also display a diverse evolution in cloud-top temperature (Fig. 6b), roughly matching the range probed by ACTIVATE remote sensing (Fig. 5) near 0 h. MERRA2-based MCAO indices along trajectories (Fig. 6c) show a collective decrease after the maximum, with cases retaining their relative strength (e.g., 11 January 2022 remains strongest throughout). MERRA2-based large-scale subsidence shows vastly different values with strong fluctuations over time, but always displaying negative median values (i.e., downward motion).*

[Figure]

**Fig. 6**: *Relative to each trajectory's time of surface flux maximum, we present statistics (i.e., median shown as lines and interquartile range shown as shading) per selected case,*

*showing GOES-16 cloud properties and MERRA-2 meteorological boundary conditions: (a) cloud cover, (b) cloud-top temperature, (c) MCAO index, and (d) $w_{700\ hPa}$.*

In contrast to the Reviewers opinion, we believe that there is value in having several samples of strong marine cold-air outbreaks (even if the selection were less diverse than shown above): it provides a more robust statistical sample of a cloud regime transition that is considered poorly understood in the community. We added the following paragraph to the Discussion (lines 381-386):

*The selected cases constitute generally strong MCAOs and should provide excellent targets for weather and climate model development. While ACTIVATE covers a wider spectrum of conditions, the selected subset forms a robust statistical sample of unique mixed-phase clouds. The range in meteorological forcing and resulting cloud cover and cloud-top temperature evolution provides a testbed for model developers to explore the inclusion of uncertain mixed-phase processes and benchmark against observational targets from ACTIVATE and satellite. The wide range in cloud-top temperature has the potential to serve as proxy of a warming climate.*

2. **Although the authors mention that users can adjust the criteria, I think some justification or more description of how the authors arrived at the selected thresholds for the parameters in Table 1 is needed. Also were any other criteria considered by the authors that might be relevant, but were not included (such as surface fluxes)?**

We agree with the Reviewer. Our metrics and their thresholds were motivated by the various objectives already described in Section 2.5. We added the following description to better connect metrics, thresholds, and objectives (lines 179-202):

*...indicate several qualities:*

(1) ***Stereotypical postfrontal conditions*** *that often emerge as MCAOs. We filter for an Eastward boundary layer wind direction ('Primarily Eastward flow') that is expected from extratropical cyclone dynamics in the postfrontal sector (e.g. Tselioudis and Grise, 2020). To obtain strong MCAOs that typically undergo faster cloud regime transitions and were in better reach of the aircraft during ACTIVATE., we also impose a MCAO index threshold ('Maximum MCAO index > 10 K'), which we also consider a proxy for elevated surface fluxes (and were therefore left out as a criterion). Typical large-scale subsidence should disallow high-level clouds that may hinder satellite retrievals; we examine GOES-16 retrievals to verify the absence of high-level clouds ('High cloud fraction < 10 %').*

(2) ***Flights that followed the MBL air mass in a quasi-Lagrangian manner with key instruments operational****. We use trajectory and flight path locations to measure spatial alignment, rewarding alignment during a sizeable fraction of the flight ('Downwind portion with respect to flight > 20 %') and additionally over a certain distance ('Downwind distance > 200 km') that we consider large enough to detect cloud property changes. We further examine if RSP, a key instrument to measure cloud micro- and macrophyiscal properties, was able to probe a sizeable portion ('Downwind RSP availability > 30 %') across this distance. Lastly, we explore the availability of Lagrangian airmass revisits, ensuring that horizontal legs are long enough to have aircraft probes collect data ('Minimum duration of Lagrangian legs > 1 minute'). We apply a similar metric to RSP as well ('Lagrangian RSP availability') to ascertain favorable potential sun-object-observer geometries.*

(3) *Where possible **liquid and frozen precipitation-sized hydrometeors that may drive a larger cloud regime transition**, as well as **elevated aerosol concentrations from new particle formation that may delay transitions**. We use data from in-situ cloud probes to indicate the presence of the various hydrometeor types ('Frozen hydrometeors measured', 'Drizzle-sized particles measured' and 'Rain-size particles measured'). We additionally survey in-situ aerosol probes for traces of new particle formation; we use condensation nucleus ratios to indicate the presence of freshly nucleated particles $3\ nm < D_p < 10\ nm$ ('Recent particle formation frequency > 5 %', e.g., Corral et al., 2022) that are expected to grow into larger sizes and drastically increased concentrations ('High aerosol concentration frequency > 5 %').*

We consider the MCAO index a proxy for surface fluxes and, therefore, left the latter one out. We included a brief note into the first item of the above list.

3. **72-74: In the context of this report, it is important to mention what these studies have already done. At a later point it should also be mentioned which cases have been studied in which of these publications. Possibly add an indication in Figure 4?**

We agree with the Reviewer and expanded Section 1 as follows (lines 80-85):

*Li et al., (2021) examined 28 February and 1 March 2020 using Eulerian LES and explored the dependence on meteorological forcing. With a focus on 1 March 2020, Chen et al. (2022) investigated the mesoscale cloud morphology in mesoscale simulations, while Tornow et al. (2022) studied aerosol dilution from FT entrainment. Seethala et al. (2024) surveyed numerous flights (i.e., 1 March 2020, 29 January 2021, 3 February 2021, 5 March 2021, and 8 March 2021) to explore mixed-phase cloud microphysical properties with distance from the coast.*

Considering separate objectives between previous work and this study, we left Figure 4 unchanged.

**Specific Comments**

1. **1, 29: The statements make it sound like marine cold-air outbreaks are purely a cloud phenomenon. Please consider revising: e.g., 'often appearing as part of [...]'**

Done.

2. **43-45: I am confused by this sentence since it seems to suggest that inhibition of vertical transport leads to the development of convection.**

Numerous studies have shown that precipitation-induced MBL stratification, reduced vertical transport (often referred to as "decoupling"), and changing cloud morphology coincide in MCAOs and other boundary layer clouds. We added the following to ll. 47-48:

*...inhibiting vertical transport (often referred to as "decoupling", e.g, Abel et al., 2017, Yamaguchi et al., 2017) ...*

3. **58: Are there any references or links for CAESAR already?**

We are not aware of any CAESER references.

4. **59: These field campaigns could be mentioned by name.**

We added specific campaigns, as suggested by the Reviewer (lines 65-68):

*..., that is (AC)$^3$ (Wendisch et al., 2023) that included AFLUX (Aircraft campaign observing FLUXes of energy and momentum in the cloudy boundary layer over polar sea ice and ocean), MOSAiC-ACA (Multidisciplinary Drifting Observatory for the Study of Arctic Climate – Airborne observations in the Central Arctic, Mech et al., 2022, Moser et al., 2023), and HALO-(AC)$^3$ (High Altitude and Long Range Research Aircraft – AC$^3$ project, Wendisch et al., 2024).*

5. **70: Is that the maximum or average MCAO index > 0 K for these 71 flights.**

We clarified in the text that we mean "maximum".

6. **Section 2.1: I am quite confused with some things here since notation does not appear to be consistent. Potential temperature is mentioned but derivatives are written as d$T$. Meridional wind speed changes are expressed as dv/dt which I interpret as the temporal change of the full wind (so it is neither the change in the meridional direction nor the change of the meridional wind). Please edit these things for clarity.**

We agree with the Reviewer and removed inconsistencies. For clarification we now show the wind vector (which contains both meridional and zonal components) as $v_h$.

7. **102: 'connect […] as lines' might be better.**

Done.

8. **131: How accurate is it to assume that all clouds are within the BL? How frequent was contamination by higher clouds and what impact did it have?**

We appreciate the Reviewer's concern and now provide a GOES16-based cloud-top height probability distribution along all trajectories (shown below and included in the appendix). 75 % of data points show clouds within the bottom 3.5 km of the atmosphere, which is expected for MCAOs. Nearly omnipresent subsidence (e.g., Fig. 3) is expected to largely disallow high-level clouds. We found 15 % to be above 5.0 km (i.e., well above the anticipated MCAOs cloud deck).

[Figure]

**Fig. A1:** *From GOES-16 cloud-top heights that we collocated along all trajectories, we produce stacked density functions for two categories.*

To consider potential implications for individual cases, we introduce another metric in Tab. 1 and display it in Fig. 5:

*High cloud fraction < 10 % - Uses Lagrangian trajectories and checks whether collocated GOES16 cloud properties contain more than 5 % of data points with cloud-top height above 5 km.*

We also expand the respective portion of the text (lines 142-144):

*While including clouds of all heights, we verify that clouds are mostly of low-level character (see Section 2.5). Fig. A1 shows ~75 % of cloud-top heights within 3.5 km of the surface and ~15 % above 5.0 km.*

and highlight the new metric in Section 3 (ll. 274-275):

*... and about half are free of high clouds.*

9. **162-165: It would be good to add to Table 1 which of the qualities mentioned here each criteria in the table corresponds to.**

Please find this point addressed in the Reviewer's major concern.

10. **182: Mention that you mean the identified front at the bottom of the image.**

Done.

11. **229: 'All flights have at least three scores' sounds confusing. Please consider revising wording: 'fulfil at least three scores'.**

Done.

12. **277-278: Both of those flights were not selected. Why if they cover most criteria and the cloud transition. I think a 10 K threshold for the MCAO index is quite high. Was the value for these two cases close to 0 K?**

Please find a brief discussion of these cases in above response to the Reviewer's first major concern. Both cases showed MCAO indices above 10 K, as marked in Fig. 4.

13. **Fig. 4: NA and TRUE have the same symbol. Further, it is quite hard to figure out the correct line for each case. Possibly, ticks on the left and a line connecting the cloud distributions to the score sheet could be added?**

**The caption should also mention what the shaded areas in the score sheet mean.**

We now show improved NA colors and ticks, and we expanded the caption. We were unable to draw lines between both halves of this figure. We now explain shaded areas in the caption.

14. **For future submissions the authors should include all figures in the main text near where they are mentioned as instructed in the ACP submission guidelines.**

Following the journal guidelines, we enabled figure placement in the main text. We presume that figure size and number led to the current organization of text and figures.

**Typographical:**

1. **73: Try to be consistent with MCAO vs CAO.** Done.
2. **228: Fig. 4** Done.
3. **312: 'favorable to certain'**

We changed to: *…ice multiplication processes … that are considered highly uncertain.*

4. **Fig. 4 caption: 'gray areas'** Done.

**Reviewer 2**

This report discusses the importance of representing marine cold-air outbreaks (MCAOs) in weather and climate models and how recent field campaigns can help improve that. It provides a summary of the undertaken flights and how they could be used for future modelling studies.

In general, I enjoyed reading the manuscript as it provides a solid overview of the flights and the functional instruments during the single flights. I do have some comments that should be addressed before publication.

**Major comments**

1. **I believe the discussion of suitable cases for Lagrangian studies should be revisited and articulated more clearly. Currently, this is to some degree confusing as it is difficult to follow the single dates. I would introduce a numbering from 1 to XX flights and reference these numbers as looking for the single dates mentioned in the text and compare it to the figures is quite cumbersome. An idea would also be to cluster the flights by the chosen criteria instead of having a chronological order.**

In response to the Reviewer's concern, we updated Figure 3 and 4 (also shown below) to now list data separated into yearly panels and insert flight numbers before flight dates. We also updated flight references throughout Sections 3.3 and 3.4.

[Figure]

[Figure]

**Overview of ACTIVATE-measured Cloud Properties**

2. **As discussed in Sect. 3.4, the selected case studies show some quite large variations with respect to cloud droplets, cloud heights, and cloud top temperature. This variety should be discussed more especially in the context of constraining weather and climate models.**

Reviewer 1 raised a similar concern - please find this concern addressed in the Reviewer 1's first major point.

**Minor comments**

1. **Line 9/10: "were better aligned with the MBL flow" – I do not understand what you mean by MBL flow**

We substituted "flow" with "wind direction" for clarification.

2. **Line 18: there must be more reference to the cloud-climate feedback than only McCoy et al, 2023**

We agree with the Reviewer and added more references for extratropical cloud-climate feedback (ll. 19-20):

*... (e.g., Frey and Kay, 2018, Zelinka et al., 2018, McCoy et al., 2019, ...*

3. **Line 35: "MBL downwind" – downwind from what?**

We clarified this in the text by inserting "farther".

4. **Line 40: I disagree with the "thereby" here coming from the previous sentence. I would turn the sentence around.**

Done.

5. **Line 51: "shaping prevalent aerosol with fetch" – what does this mean?**

We refer to processes that may modify the aerosol's number and mass and now clarify this in the text (ll. 54-55):

*...modifying aerosol size and mass composition with fetch*

6. **Line 50 – 61: I think it could be beneficial to have an overview figure with all campaigns marked and highlighting the measurement area of ACTIVATE**

While we generally agree with the Reviewer, we think it is outside the scope of this work and better suited inside a review paper.

7. **Line 88 – 90: can you split up this long sentence in two?**

Done.

8. **Line 105: "at time steps around the time of interest" – be more specific**

We specified this as follows (l. 116):

*...three hours before and after the time of interest.*

9. **Line 114: "which we assume is representative of MBL" – add references or more justification why this is applicable**

We added a reference to the work by Seethala et al. (2021).

10. **Line 133: "load" – please change the verb**

We changed it to "collect".

11. **Line 145: use CCN as abbreviation as you have already introduced it**

We introduce "CN" to abbreviate "condensation nuclei" (please note the difference compared to CCN).

12. **Line 166-168: I think the criteria in Table 1 need to be more clearly introduced, as I was surprised and confused by this paragraph, and did not think it was helpful. A proper discussion of Table 1 would be good.**

Reviewer 1 raised a similar concern – please see our response to Reviewer 1's second concern.

13. **Line 176-180: switch around the two criteria to be consistent with the explanation in Section 2.1 and give a paragraph title, e.g., "1. Hewson (1998): searching for spatial ... "**

Done.

14. **Line 188 – 193: split up this single sentence**

Done.

15. **Line 204: orange should be grey?**

Correct. We changed this in the text.

16. **Line 209: "and MCAO indices" – you already said that in the beginning of the sentence**

We believe that MCAO strength can be measured in several ways (e.g., via surface fluxes or the MCAO index). For completeness, we prefer retaining the current list.

17. **Line 210: give some more background information to the turbulent surface fluxes**

We expanded the text as follows (ll. 249-250):

*..., where latter is expected to increase with near-surface wind speed and surface-air differential in temperature and humidity.*

18. **Line 212: As far as I can see, the data population is very sparse for the appended figures – can you add the number of data points going into your box whiskers, and discuss that as well? For case late 2020, early 2021 it looks like only 3 or 4 flights are selected, which are not enough to have box whiskers for that.**

We now added sample sizes to the legend of Fig. 3 and Fig. A3 (now Fig. A2) and include the following text (l. 252):

*Note that sample size is smaller in Fig. A2, preventing box-whisker plots from reliably producing whiskers.*

19. **Line 219: what do you mean by process signatures?**

We clarified this in the text as follows (ll. 260-261):

*(i.e., sufficient distance to detect process signatures, such as CCN decrease from precipitation formation)*

20. **Line 222: remove that sentence "Additionally, we indicate whether flights capture"**

In this text, we give less importance to criteria 5 and 6 and prefer listing these separately from criteria 1 through 4.

21. **Line 226: While I do understand that some work is still in progress, here a bit more explanation is required how this is done for new particle formation**

We included an additional reference (Zheng et al., 2021) and expanded the text as follows (l. 267)

*...that undergoes particle growth and could result in...*

22. **Line 234: "not shown" is not a good practice anymore remove and add to appendix**

We agree with the Reviewer. In this case we referenced preliminary analysis that merely shows that most flight tracks indeed cross an airmass twice. Given that many flights show non-linear paths, such insight is unsurprising (e.g., traveling on a triangular shape, one is guaranteed to intercept an airmass twice). Instead of adding a plot that demonstrate the obvious, we suggest to rephrase this portion as follows (ll. 277-278):

*While air mass revisits are nearly unavoidable on mostly non-linear flights tracks, ...*

23. **Line 265: I believe Figure A1 was never referenced, please double-check**

We now reference Fig. A1 (now Fig. A5) in newly added Section 3.5.

24. **Line 268-269: I do not understand this part with where the aircraft was located downwind and farther downwind.**

We expanded the text as follows (ll. 315-318):

*...downwind of an event (e.g., in clouds that increased beyond 75 % past cloud formation)...*

*...an event was located farther downwind than seen by aircraft (e.g., in clouds that have yet to increase towards 75 % cloud cover for cloud formation).*

25. **Line 307: I expect there to be also other references for the CCN dilution in the MBL – please add.**

We are not aware of other references.

**Technical comments**

1. **Line 19: "contributes" to contribute** Done.
2. **Line 65: wrong citation style *(\citet{} vs \citep{})* ◊ this also goes for other instances: line 69, 105, 157** Done.
3. **Line 73/161: stay consistent with MCAOs and not CAOs** Done.
4. **Line 110: degree sign needs to go after the numerical value** Done.
5. **Line 127: space between < and 6.0** Done.
6. **Line 228: I believe the wrong figure is referenced here.**

We now reference Fig. 4.

7. **Line 240: space between number and unit (%)** Done.

8. **Line 312: remove space between 9 and 5** Done.

9. **Figure 1: please check that all colors are also explained in the figure caption, e.g., the yellow line? Dashed and solid lines? Green lines?**

We expanded the caption of Fig. 1 as follows:

*...flight track (yellow line)...ocean surface (country borders shown in green lines)...*

10. **Figure 2: why did you choose vertical velocities at 700 hPa? Please discuss**

We choose 700 hPa as a level typical for the free troposphere. We added a brief explanation to the caption of Fig. 2:

*..., where 700 hPa is expected to represent free tropospheric conditions.*

11. **Table 1: check that there is spacing between numerical values and units** Done.

12. **Figure 3: MCAO index subplot could benefit from a dashed black line at 0 (as in $w_{700\,hPa}$) for readability; the box whiskers should be in the same order as the labels, ideally actually blue, grey and then orange.** Done

13. **Figure 4: As said above, I believe a proper numbering of the flights would be useful, instead of the single dates. Please also explain the orange and blue shading in the score sheet. It is very hard to distinguish TRUE and NA – can you adapt that, such that maybe NA is represented by a cross or triangle? I would also color the selected flights labels and not only the score sheet. I would also sort the flight chronologically from top to down. There is a missing space in the figure caption in "gray areas". The numbers in the legend are not needed.**

As addressed in the major point, we added numbering and separated by years. We also now explain shading, changed NA value colors, and removed numbers in the legend.

14. **Figure 5: similar comments as for figure 4.**

As addressed in the major point, we added numbering and clustered points.

15. **Figure A1: not referenced?**

Addressed in a previous concern.

16. **Figure A2: can you choose colors for cloud formation and breakup that are consistent with Figure 4? Also red and green is not color vision deficiency friendly. Please avoid that combination for line plots.**

Done.

17. **Figure A3: as mentioned above, I would like to have the number of data points going into the box whiskers. Especially the late 2020, early 2021 case seems like very few flights which do not warrant a box plot – please discuss that. Further, when data is missing such as for late 2020, early 2021 a connecting line should be not plotted. Please adjust that.**

Done. The smallest number of data points is four. We now point out in the caption in Fig. A3 (now Fig. A2) that no whiskers are too be expected.

18. **Figure A4: have a caption that fully explains the figure.**

We expanded the caption as follows:

*Similar to Fig. 1, but here shown for the first flight of each selected day (yellow) with the trajectory (red) that maximally connects both flights, determined through the maximum fraction of timestamps within 20 km and 1 hour of each trajectory.*

**References**

Frey, W. R., & Kay, J. E. (2018). The influence of extratropical cloud phase and amount feedbacks on climate sensitivity. *Climate dynamics*, *50*, 3097-3116.

McCoy, D. T., Field, P. R., Elsaesser, G. S., Bodas-Salcedo, A., Kahn, B. H., Zelinka, M. D., ... & Wilkinson, J. (2019). Cloud feedbacks in extratropical cyclones: insight from long-term satellite data and high-resolution global simulations. *Atmospheric Chemistry and Physics*, *19*(2), 1147-1172.

Zelinka, M. D., Klein, S. A., Qin, Y., & Myers, T. A. (2022). Evaluating climate models' cloud feedbacks against expert judgment. *Journal of Geophysical Research: Atmospheres*, *127*(2), e2021JD035198.

Zheng, G., Wang, Y., Wood, R., Jensen, M. P., Kuang, C., McCoy, I. L., ... & Wang, J. (2021). New particle formation in the remote marine boundary layer. *Nature communications*, *12*(1), 527.